# Comparative genome analysis of *Salmonella enterica* serovar Gallinarum biovars Pullorum and Gallinarum decodes strain specific genes

**Rajesh Kumar Vaid**[1]*, **Zoozeal Thakur**[1], **Taruna Anand**[1], **Sanjay Kumar**[2], **Bhupendra Nath Tripathi**[2]

1 Bacteriology Laboratory, National Centre for Veterinary Type Cultures, ICAR-National Research Centre on Equines, Hisar, Haryana, India, 2 Bacteriology Laboratory, ICAR-National Research Centre on Equines, Hisar, Haryana, India

* rk_vaid@yahoo.com

**Data Availability Statement:** All relevant data are within the manuscript and its Supporting Information files.

## Abstract

*Salmonella enterica* serovar Gallinarum biovar Pullorum (bvP) and biovar Gallinarum (bvG) are the etiological agents of pullorum disease (PD) and fowl typhoid (FT) respectively, which cause huge economic losses to poultry industry especially in developing countries including India. Vaccination and biosecurity measures are currently being employed to control and reduce the *S*. Gallinarum infections. High endemicity, poor implementation of hygiene and lack of effective vaccines pose challenges in prevention and control of disease in intensively maintained poultry flocks. Comparative genome analysis unravels similarities and dissimilarities thus facilitating identification of genomic features that aids in pathogenesis, niche adaptation and in tracing of evolutionary history. The present investigation was carried out to assess the genotypic differences amongst *S.enterica* serovar Gallinarum strains including Indian strain *S*. Gallinarum Sal40 VTCCBAA614. The comparative genome analysis revealed an open pan-genome consisting of 5091 coding sequence (CDS) with 3270 CDS belonging to core-genome, 1254 CDS to dispensable genome and strain specific genes *i.e.* singletons ranging from 3 to 102 amongst the analyzed strains. Moreover, the investigated strains exhibited diversity in genomic features such as virulence factors, genomic islands, prophage regions, toxin-antitoxin cassettes, and acquired antimicrobial resistance genes. Core genome identified in the study can give important leads in the direction of design of rapid and reliable diagnostics, and vaccine design for effective infection control as well as eradication. Additionally, the identified genetic differences among the *S. enterica* serovar Gallinarum strains could be used for bacterial typing, structure based inhibitor development by future experimental investigations on the data generated.

## 1. Introduction

Pullorum disease (PD) and fowl typhoid (FT) are two distinct septicaemic diseases caused by non-motile *Salmonella enterica* subsp. *enterica* serovar Gallinarum (*S*. Gallinarum) biovar

**Funding:** This work was partly supported by Department of Biotechnology, Govt. of India, New Delhi (Garnt No. BT/BI/25/07/2012-BIF) (URL: http://dbtindia.gov.in/) award to RKV and Director, National Research Centre on Equines, National Centre for Veterinary Type Cultures, Hisar, Haryana, India. ZT acknowledges DBT "Bioinformatics infrastructure facility (BIF)" for providing Research Associateship. Funding source (s) had no involvement if any, in study design; in the collection, analysis and interpretation of data; in the writing of the report; and in the article preparation or decision to submit the article for publication.

**Competing interests:** The authors have declared that no competing interests exist.

Pullorum (bvP) and biovar Gallinarum (bvG), respectively, which exhibit host-specificity towards poultry and aquatic birds [1–3]. The diseases caused by these invasive avian pathogens cause high morbidity and acute mortality in poultry in India and various countries of Asia, Africa and South America. Pullorum disease (PD) occurs in young birds and persists for long periods in spleen and reproductive tract which leads to high mortality [4]. The disease is characterised by white diarrhoea chaperoned with increased antimicrobial resistance (AMR) and high infection rates [5, 6]. On the other hand, FT can affect birds of all ages but primarily occurs in adult birds and results in variable morbidity depending on the age, species, and breed of the bird [7]. *Salmonella* Enteritidis, a major food-borne pathogen causes infection which leads to enteritis in multiple hosts in addition to poultry [8]. Although *S.* Gallinarum has negligible importance in humans, *S.* Enteritidis is an important zoonoses [8, 9]. The *S.* Gallinarum, in addition has been found to be a recently evolved ancestors of *S.* Enteritidis [10].

The virulence of *Salmonella* spp. is mediated by an arsenal of genes which are capable of invasion, replication, and colonization inside the host cells [11, 12]. The genetic factors involved in pathogenesis of FT and PD at molecular and cellular mechanisms are still under elucidation [2, 13]. The genomic sources of virulence systems in *Salmonella enterica* serovars are mainly divided into two elements, one of which is horizontally acquired chromosomally located mobile genetic elements known as *Salmonella* pathogenicity Islands (SPIs), and prophage elements, whereas other is plasmids associated with salmonellae [14]. These genetic elements have played a seminal role in various ecological niche adaptations in different host and pathogenicity life-style in *Salmonella* by encoding for proteins carrying out cell-adherence, cellular invasion at host level, induction of innate immune and/or pathophysiological responses to infection [15, 16]. Two pathogenicity islands, *Salmonella* pathogenicity island 1 (SPI-1) and SPI-2 have been described, which play roles in mediating disease by *Salmonella enterica* through their respective type III secretion systems (TTSS). SPI which encode T6SSs have been detected in SPI6, SPI19, 20, 21 in *S.* Gallinarum and *S.* Enteritidis [17].

Additionally, prophage lysogeny is a rich contributor of *Salmonella* genome diversity, and their acquisition leads to enhanced virulence and pathogenicity [18, 19]. The prophage elements are being increasingly utilised as molecular markers for strain discrimination [20, 21]. Another crucial genetic element that is involved in persistence, virulence and AMR includes toxin-antitoxin system (TA system) [22–24]. Furthermore, the *Salmonella* serovars including the Gallinarum and Enteritidis are increasingly reporting incidence of antimicrobial resistance (AMR) [25, 26]. The comparative genomic analysis of pathogens helps in elucidating the diversity and depth of functional set of genes in form of pan-genome and decoding of evolutionary history [27, 28]. Moreover, it aids in functional annotation, as well as decodes molecular mechanisms underlying pathogenesis and niche adaptation by precise measurement of genetic variation within and between pathogenic groups [29–31].

Recently we determined the first draft genome sequence of an Indian strain of *S.* Gallinarum VTCCBAA614 isolated from diseased poultry [32]. The present investigation dealt with whole genome characterization and assessment of genotypic differences amongst bvG and bvP strains including *in house S.* Gallinarum Sal40 VTCCBAA614 strain (Table 1). The study unravelled pan-genome, core-genome, dispensable genome, and strain specific genes of the analyzed *S.* Gallinarum strains. Moreover, the investigation decoded diversity in virulence factors, genomic islands, prophage regions, TA systems, and acquired AMR genes in the investigated strains via employment of various computational tools. The genotypic differences revealed by the study will serve as the genomic resource for the identification and discrimination of biovars, and could form the basis of structure based inhibitors design and that would be beneficial to poultry industry. Although comparative genomics analysis of host-adapted salmonellae genomes have elucidated various evolutionary and virulence themes [5, 8], this is the

**Table 1. Genome statistical information of nine investigated *Salmonella* strains as obtained by Prokka pipeline and ANI calculator.**

| #Organism name | Geographic location | Accession number | Assembly status | Total length | Size (MB) | GC Content % | CDS | tRNA | tmRNA | rRNA | rr | SNP detected |
|---|---|---|---|---|---|---|---|---|---|---|---|---|
| *S. enterica* Enteritidis str. P125109 | United Kingdom | NC_011294.1 | Complete | 4,685,848 | 4.68 | 52.17 | 4352 | 85 | 1 | 22 | 2 | - |
| *S. enterica* Gallinarum str. 287/91 | Brazil | NC_011274.1 | Complete | 4,658,697 | 4.65 | 52.20 | 4452 | 77 | 1 | 22 | 2 | 7832 |
| *S. enterica* Gallinarum str. 9184 | Not available | NZ_CP019035.1 | Complete | 4,609,911 | 4.60 | 52.20 | 4399 | 78 | 1 | 22 | 2 | 7505 |
| *S.enterica* Gallinarum str. Sal40 | India | JSWQ00000000.1 | Contig | 4,598,206 | 4.59 | 52.22 | 4455 | 68 | 1 | 4 | 2 | 7946 |
| *S.enterica* Pullorum str. ATCC 9120 | USA | NZ_CP012347.1 | Complete | 4,694,842 | 4.69 | 52.19 | 4474 | 81 | 1 | 23 | 2 | 7424 |
| *S.enterica* Pullorum str. S06004 | China | NC_021984.1 | Complete | 4,682,599 | 4.68 | 52.14 | 4635 | 80 | 1 | 22 | - | 12726 |
| *S.enterica* Pullorum str. QJ-2D-Sal | China | NZ_CP022963.1 | Complete | 4,728,875 | 4.72 | 52.17 | 4548 | 75 | 1 | 22 | 2 | 7543 |
| *S.enterica* Gallinarum/ Pullorum str. CDC1983-67 | China | NC_022221.1 | Complete | 4,623,089 | 4.62 | 52.23 | 4397 | 78 | 1 | 22 | 2 | 7786 |
| *S.enterica* Gallinarum/ pullorum str. RKS5078 | Not available | NC_016831.1 | Complete | 4,637,962 | 4.63 | 52.21 | 4451 | 75 | 1 | 22 | 2 | 8038 |

first report for the assessment of genotypic differences, which includes an Indian strain among the investigated strains from America and China.

## 2. Material and methods

### 2.1 Data collection of *Salmonella* genomes

The complete genome sequences of eight *S.* Gallinarum strains *i.e.*, *S.* Gallinarum str. 287/91 (NC_011274.1), *S.* Gallinarum str. 9184 (NZ_CP019035.1), *S.* Pullorum str. ATCC 9120 (NZ_CP012347.1), *S.* Pullorum str. S06004 (NC_021984.1), *S.* Pullorum QJ-2D-Sal (NZ_CP022963.1), *S.* Gallinarum/pullorum str. CDC1983-67 (NC_022221.1), and *S.* Gallinarum/pullorum str. RKS5078 (NC_016831.1) including draft genome sequence of *S.* Gallinarum Strain VTCCBAA614 (isolated from chicken in India from FT inflicted broiler flock) were mined from NCBI database to decode genotypic differences among the *S. enterica* serovar Gallinarum strains. In addition the genome sequence of *S.* Enteritidis str. P125109 (NC_011294.1) [10] a well studied pathogenic *Salmonella* strain was also retrieved from NCBI for employment as reference for comparative genome analysis (Table 1). Sequence quality of genome assemblies is dependent on the sequencing technology used, genome coverage and aim of the sequencing. Information regarding the investigated genomes such as sequencing technology used, genome coverage, as well as assembly method was mined from literature as well as NCBI database and is listed in S2 Table. In addition, measures of genome quality such as completeness, contamination, coarse consistency, and fine consistency as identified by EvalG and EvalCon tools of PATRIC database are also listed (S2 Table)

The Indian strain *S.* Gallinarum VTCCBAA614 [32] is available from our culture collection at NCVTC, NRCE Hisar. All the sequences were extracted in fasta and GenBank (gbk) format.

## 2.2 Genome statistics and visualization

Genomic features such as number of CDS, tRNA, tmRNA, rRNA and repeat region (rr) were determined by running Prokka pipeline at UseGalaxy webinterface (Table 1) [33]. The GC% of the genomes was determined by ANI Calculator which uses OrthoANIu algorithm [34]. Genomes were aligned with *S*. Enteritidis P125109 genome taken as reference genome. Alignment and single nucleotide polymorphism (SNP) detection was performed by using nucmer and show-snps (with Clr parameter) components of MUMmer package V 3.1 [35]. In addition, circular plot visualization of CDS, GC content, and GC skew with *S*. Enteritidis str. P125109 as reference genome against other *Salmonella* genomes was generated by EDGAR [36].

## 2.3 Pan- and core-genome calculations

Pan-genome is the complete set of orthologous genes (OGGs) harboured within a collection of investigated genomes, whereas, the core-genome refers to the set of genes present in all the genomes of a collection. On the other hand, dispensable genome refers to the set of genes harboured by one or a subset of investigated genomes. Singleton genes are the unique genes that do not have any homologs in the investigated genomes [37, 38]. EDGAR served as the resource for pan-genome, core-genome, and singleton genes calculations for the selected *Salmonella* strains [36]. A customised project was set up by EDGAR to conduct all the calculations for pan-genome, core-genome, dispensable genome, and singletons with selection of *S*. Enteritidis str. P125109 as reference genome.

## 2.4 Functional annotation of pan- and core-genome

The functional annotation of core-genome and complete set of strain-specific genes via orthology assignment was carried by eggNOG-mapper v2 online portal (http://eggnog-mapper.embl.de) [39, 40]. The tool employs precomputed clusters and phylogenies from its *in house* database and provides orthology assignment to a large set of sequences via fast orthology mapping. Out of the total 29,430 CDS that comprised core-genome of nine investigated *Salmonella* genomes, 29,125 (98.96%) CDS were queried by eggNOG-mapper v2 for orthology mapping. On the other side, 153 (48.11%) out of the total 318 strain specific CDS (singletons) were queried for orthology mapping.

## 2.5 Phylogenetic analysis of the strains

The identified core-genome of the investigated *Salmonella* strains was employed by EDGAR to decipher phylogenetic relationship among the strains. Alignment of each CDS set was obtained by MUSCLE [41], and then further joined to form one huge alignment. The FastTree software was used to construct the tree. Program employs Shimodaira-Hasegawa (SH) branch support values to verify the tree topology [42]. Conservation of gene order and genome rearrangements among *Salmonella* genomes were also explored by using EDGAR wherein *S*. Enteritidis P125109 was chosen as a reference to create synteny plots. The Indian strain, bvG VTCCBAA614, was not included, as the use of draft genomes is not recommended in synteny plot representation.

## 2.6 Detection of virulence factors

VFanalyzer tool available at VFDB (virulence factor database) was utilized to detect putative virulence factors in the investigated *Salmonella* genomes [43]. The first step in the detection of virulence factors by the tool involves construction of orthologous groups within the query genome sequence as well as in the pre-investigated *Salmonella* reference genomes archived in

the database to avoid detection of paralogs. Next, it carries iterative and exhaustive sequence similarity search against the VFDB database for accurate detection of virulence factors. Finally, a context-based data refinement process is performed for virulence factors encoded by gene clusters [43]. We also carried out additional NCBI BLAST searches for the virulence genes that showed differential distribution after manual curation of VFanalyzer tool results.

### 2.7 Detection of *Salmonella* pathogenicity islands (SPI)

Two approaches were employed to detect SPI's -the major determinant of *S. enterica* virulence via usage of web tool SPIFinder 1.0 available at (https://cge.cbs.dtu.dk/services/SPIFinder) [44] with default parameters, and NCBI BLAST search of known SPIs extracted from PAIDB v2.0 (http://www.paidb.re.kr/) against investigated genomes [45, 46].

### 2.8 Identification and *in silico* characterization of prophage sequences

Potential prophage sequences within the *Salmonella* genomes were detected and annotated by employing PHASTER (Phage Search Tool Enhanced Release). PHASTER, an improved version of PHAST phage search tools detects candidate prophage regions in the bacterial genomes and then classifies the identified regions into three classes *i.e.* intact, incomplete and questionable on the basis score obtained [47, 48].

### 2.9 Detection and analysis of Type II Toxin-Antitoxin (TA) gene cassettes

Type II TA loci were predicted in the *Salmonella* genomes by employing TAfinder [49]. The parameters used for the detection were BLAST e value: 0.01; HMMer E-value: 1; Maximum length for candidate toxin/antitoxin: 300; maximum overlap between candidate toxin and antitoxin: (-20–150).

### 2.10 Screening of acquired antimicrobial resistance (AMR) genes

Resfinder available at (https://cge.cbs.dtu.dk/services/ResFinder/) was employed to investigate the presence of acquired antimicrobial resistance (AMR) genes in the selected *Salmonella* strains [50]. Resfinder identifies acquired antibiotic resistance genes foraminoglycoside, β-lactam, colistin, fluoroquinolone, fosfomycin, fusidic acid, glycopeptide, MLS-macrolide, lincosamide and streptogramin B nitroimidazole, oxazolidinone, phenicol, rifampicin, sulphonamide, tetracycline and trimethoprim. Moreover, both known and unknown chromosomal point mutations were also detected in AMR resistance genes such as *gyr*A, *par*E, *pmr*A, *pmr*B, *par*C and 16S_*rrs*D.

## 3. Results

### 3.1 Comparative genome statistics

Comparative genome analysis of selected nine strains of *Salmonella* species (Table 1) was carried out by employing various bioinformatics tools. The average genome size of the investigated *Salmonella* strains was 4,657,781 bp, ranging from 4,598,206 bp (*S*. Gallinarum Sal40 strain VTCCBAA614) to 4,728,875 bp (*S*. Pullorum QJ-2D-Sal). The average GC content of all the analyzed genomes was 52.19%, ranging from 52.14% (*S*. Pullorum str. S06004) to 52.23% (*S*. Gallinarum/pullorum str. CDC1983-67) (Table 1). Genomic features of the investigated *Salmonella* strains were determined by running Prokka pipeline at Use Galaxy webinterface. The average number of CDS was observed to be 4463 with highest number detected in *S*. Pullorum str. S06004 (4635) and the lowest in *S*. Enteritidis str.

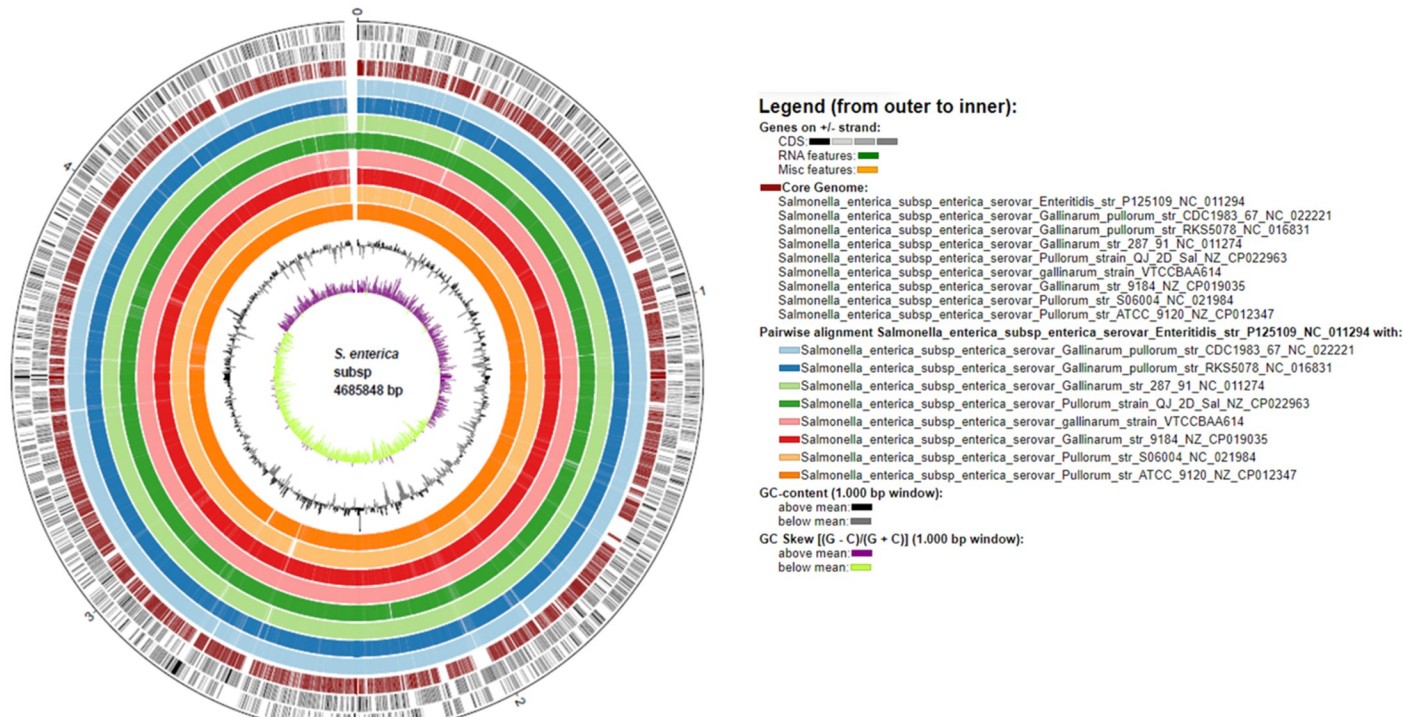

**Fig 1.** Circular plot genome representation of nine investigated *Salmonella* strains *i.e.* a) *S.* Gallinarum str. 287/91 b) *S.* Gallinarum str. 9184 b) *S.* Gallinarum Sal40 strain d) *S.* Pullorum str. ATCC 9120 e) *S.* Pullorum str. S06004 f) *S.* Pullorum QJ-2D-Sal g) *S.* Gallinarum/Pullorum str. CDC1983-67 and h) *S.* Gallinarum/Pullorum str. RKS5078 with demonstration of GC content, GC skew and CDS in reference to *S.* Enteritidis str. P125109.

P125109 (4352) (Table 1). The tRNA sequences identified by Prokka pipeline in the *S.* Gallinarum genomes fell in the range of 75–81 *i.e.* 75 in *S.* Gallinarum/pullorum str. RKS5078 and 81 in S. Pullorum ATCC 9120. It was 85 in *S.* Enteritidis str. P125109 (Table 1). In addition, all the investigated *Salmonella* genomes were detected to possess only one tmRNA. Notably, only four rRNA sequence were detected in *S.* Gallinarum Sal40 strain VTCCBAA614 in comparison to possession of 23 rRNA sequences in *S.* Pullorum str. ATCC 9120 and presence of 22 rRNA sequences in rest of the genomes (Table 1). The circular plot visualization of investigated genomes with reference to *S.* Enteritidis str. P125109 depicts varied GC content and GC skew along with predominant similarity of core-regions of the investigated genomes (Fig 1).

## 3.2 Pan and core-genome analysis unravels strain specific genes

The pan-genome calculation by customised set up in EDGAR led us to have discrete idea of the total genetic repository of the *Salmonella* genomes under study. The pan-genome of *S.* Gallinarum was found to be in open state with growth exponent value of 0.089 (95% confidence interval 0.084 to 0.093) (S1 Table). The pan-genome development plot showed steady growth with addition of each new genome and reached 5091 on addition of ninth genome which is nearly 1.1 times the average number of genes of nine strains (Fig 2A, S1 Table). On the other hand, core-genome development plot became limited to 3270 genes, with shared genes decreasing with new genome addition. The extrapolated core-genome size was 2684 (95% confidence interval 2507.66 to 2861.11) (Fig 2B, S1 Table). In addition, singleton development plot also indicated *S. enterica* serovar Gallinarum to be in an open pan-genome state as 43 new genes were predicted to be found at every genome addition (Fig 2C, S1 Table).

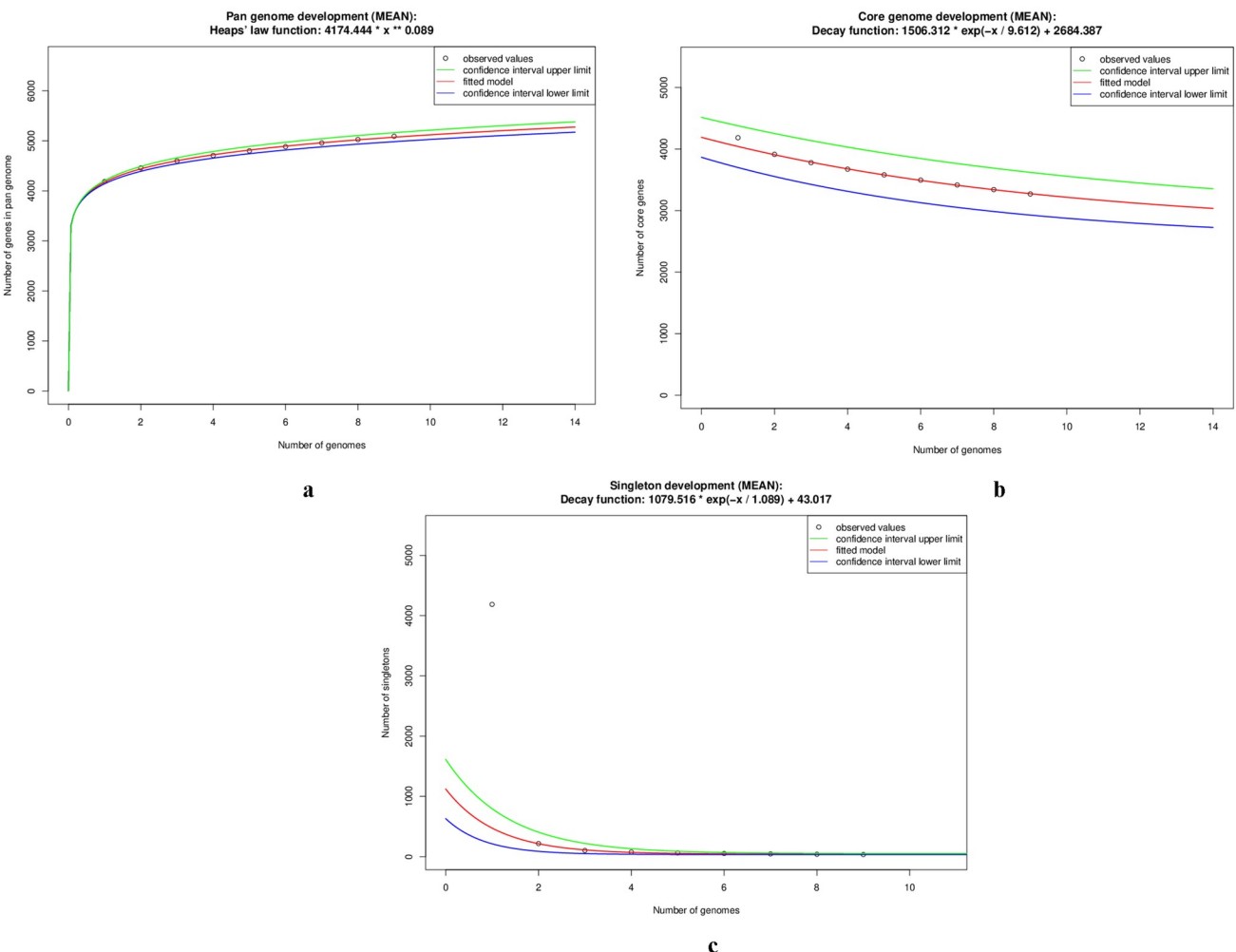

**Fig 2. Core genome development plot for nine *Salmonella* genomes.** The red curve shows the fitted model. Green represents confidence interval upper limit and blue demonstrates confidence interval lower limit.

The pan-genome of investigated *Salmonella* strains with the reference to *S.* Enteritidis str. P125109 strain is composed of 5,091 coding sequences (CDS) which include a core-genome of 3,270 (64.2%) CDS, a dispensable genome of 1254 (24.6%) CDS, and 567 (11.1%) singletons. The complete list of CDS detected as part of pan-genome, core-genome, and singletons with their function are listed in S3, S4 and S5A–S5I Tables. Notably, a total of 318 CDS in the range of 3–102 (Table 2, S5 Table) were found as strain specific genes aka singletons in the investigated genomes. Significantly, *S.* Gallinarum Sal40 strain VTCCBAA614 harboured the highest number (102) of singletons among the investigated *Salmonella* strains (Table 2, S5 Table). The singletons detected were identified to be as hypothetical proteins (61), transposase (13), membrane proteins (8), ATP binding proteins (3) among others. Interestingly, genomes of *S.* Enteritidis str. P125109 and *S.* Pullorum QJ-2D-Sal possessed 96 and 85 singletons each, respectively.

On the other hand, *S.* Pullorum str. ATCC 9120 and *S.* Pullorum str. S06004 genomes harboured ten singletons each, whereas, the rest of the genomes of possessed ≤5 singletons each (Table 2, S5A–S5I Table).

**Table 2. Number of CDS identified in pan genome, core genome and singletons of the investigated *Salmonella* genomes.**

| Organism and strain | Accession number | Pan genome | Core genome | No of singletons |
|---|---|---|---|---|
| *S.enterica* Enteritidis str. P125109 | NC_011294.1 | 5091 | 3270 | 96 |
| *S. enterica* Gallinarum str. 287/91 | NC_011274.1 | | | 3 |
| *S.enterica* Gallinarum str. 9184 | NZ_CP019035.1 | | | 4 |
| *S. enterica* Gallinarum str. Sal40 strain | JSWQ00000000.1 | | | 102 |
| *S.enterica* Pullorum str. ATCC 9120 | NZ_CP012347.1 | | | 10 |
| *S. enterica* Pullorum str. S06004 | NC_021984.1 | | | 10 |
| *S. enterica* Pullorum str. QJ-2D-Sal | NZ_CP022963.1 | | | 85 |
| *S. enterica* Gallinarum/pullorum str. CDC1983-67 | NC_022221.1 | | | 3 |
| *S. enterica* Gallinarum/pullorum str. RKS5078 | NC_016831.1 | | | 5 |

Functional annotation of core-genome and strain specific CDS by assignment of orthology (COG) was performed by eggNOG-mapper v2. The annotation by eggNOG-mapper provided detailed description of GO term, EC number, annotation level, COG category and description (S6 Table). 29,125 CDS (98.96%) out of the total 29,430 (core-genome sequences) were queried by eggNOG-mapper v2 for orthology mapping. The highest number among them belonged to the functional classes of function unknown (5923) transcription (2592), amino acid metabolism and transport (2196), energy production and conversion (2178), and cell wall/membrane/envelop biogenesis (2133) (Fig 3A, S6A Table). On the other hand, 153 out of the 318 CDS (complete set of identified singleton sequences) were queried by eggNOG-mapper v2. The highest number amongst them belonged to the classes of function unknown (68), replication and repair (24) and defence mechanism (6) among others. Thirty five singletons were not provided any COG category by eggNOG-mapper (Fig 3B, S6B Table)).

### 3.3 Phylogenetic and synteny plot analysis

A phylogenetic tree was constructed on the basis of core-genome comprising of 9,83,299 aa-residues/bp per genome and 88,49,691 in total of nine *Salmonella* genomes by EDGAR (Fig 4) using *S. enterica* serotype Enteritidis strain P125109 as a reference sequence to demonstrate the evolutionary relationship among the *S.* Gallinarum strains used in the study.SH support values for our tree were very good in general, with a minimum value of 0.772 and only two values below the maximum of 1.00. The bvG and bvP formed two distinct and strongly supported clades in the phylogeny (Fig 4). In concordance with their taxonomic classification bvG strains clustered together *i.e.* *S.* Gallinarum str. 287/91, *S.* Gallinarum str. 9184 and Gallinarum Sal40 strain VTCCBAA614) and formed one clade. Whereas, the rest of the bvP strains bifurcated into 2 clades (Fig 4).

The synteny plot analysis performed by EDGAR of the investigated *Salmonella* genomes in reference to *S.* Enteritidis str. P125109 depicted large scale genomic rearrangements which included relocations, inversions, duplications and deletions (Fig 5). It was observed that all bvP strains genomes showed high degree of conservation of gene order among genomes. The highly similar genomic rearrangement within bvP comprised of inversion and duplication. However, the bvG strain 9184 was characterized by large region of inversion, which was not observed in bvG 287/91 in reference to *S.* Enteritidis str. P125109 strain. Synteny plot of individual investigated strains against the reference genome of *S.* Enteritidis str. P125109 is shown in S1 Fig.

### 3.4 Virulence factor detection

Putative virulence factors were detected by VFanalyzer tool in investigated *Salmonella* genomes. The tool identified candidate virulence factors related to capsule, fimbrial adherence,

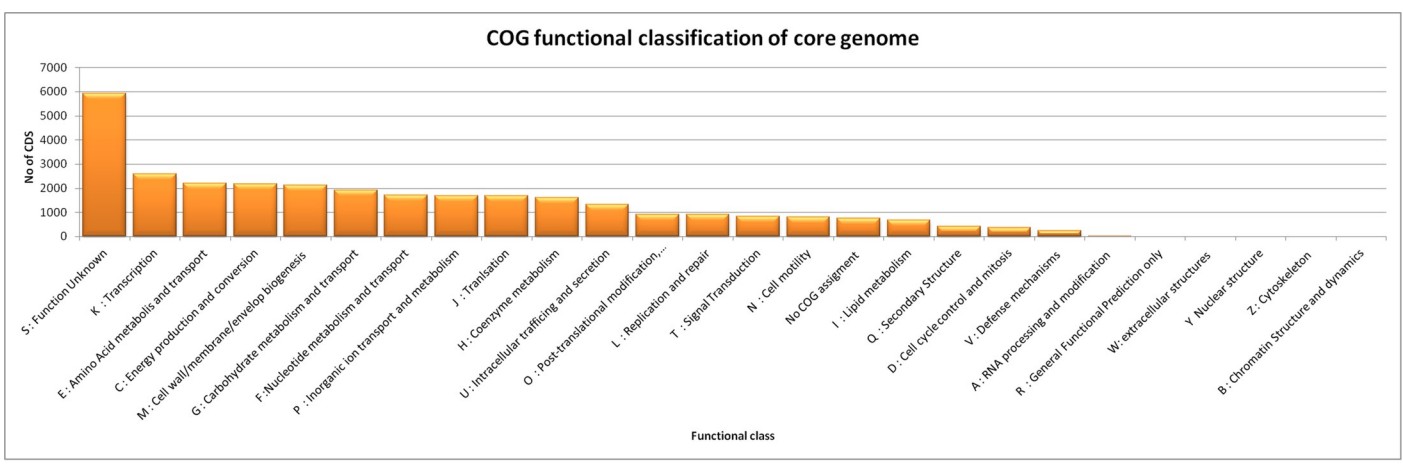

a

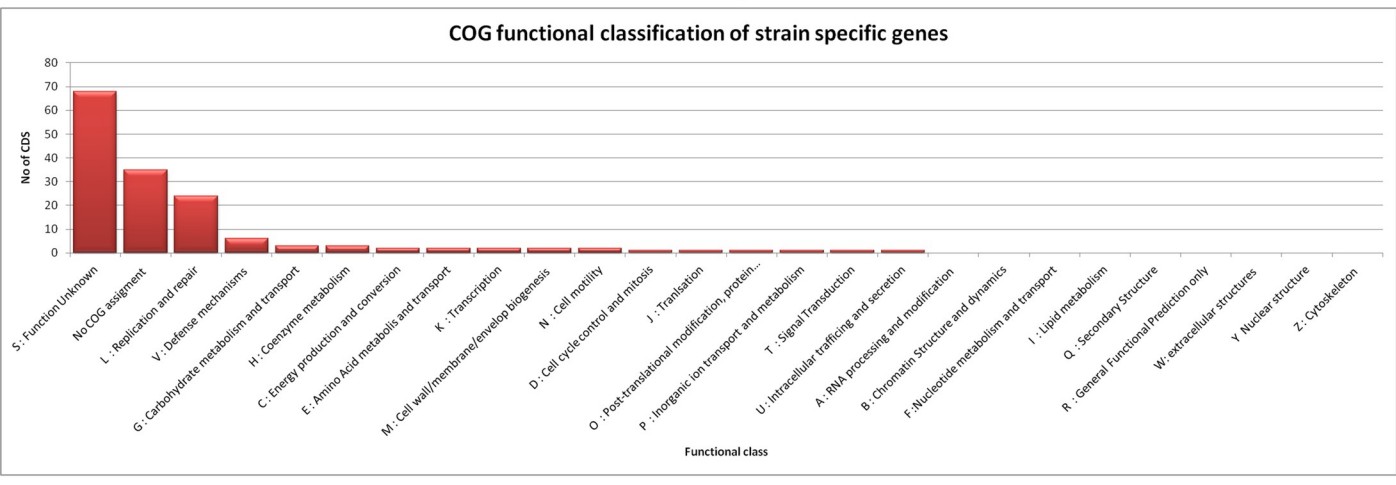

b

**Fig 3.** a) Distribution of COG functional classes in core genome of investigated *Salmonella* strains by eggnog mapper b) Distribution of COG functional categories in strain specific genes by eggnog mapper.

macrophage inducible genes, magnesium uptake, non-fimbrial adherence, regulation, secretion system, stress adaptation, autotransporter and invasion as listed (S7A–S7C Table). Fimbrial adherence genes which includes *agf/csg, bcf, fim, lpf, peg, pef, saf, sef, sta, stb, stc, std, ste,*

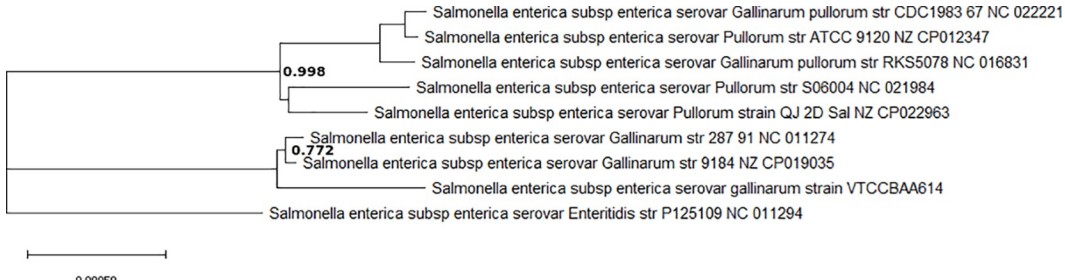

**Fig 4. Unrooted phylogenetic tree showing taxonomic positions of the investigated *Salmonella* strains constructed by EDGAR on the basis of core genome set.** Branch support values that were lower than the maximum values of 1.0 are shown at the respective branches.

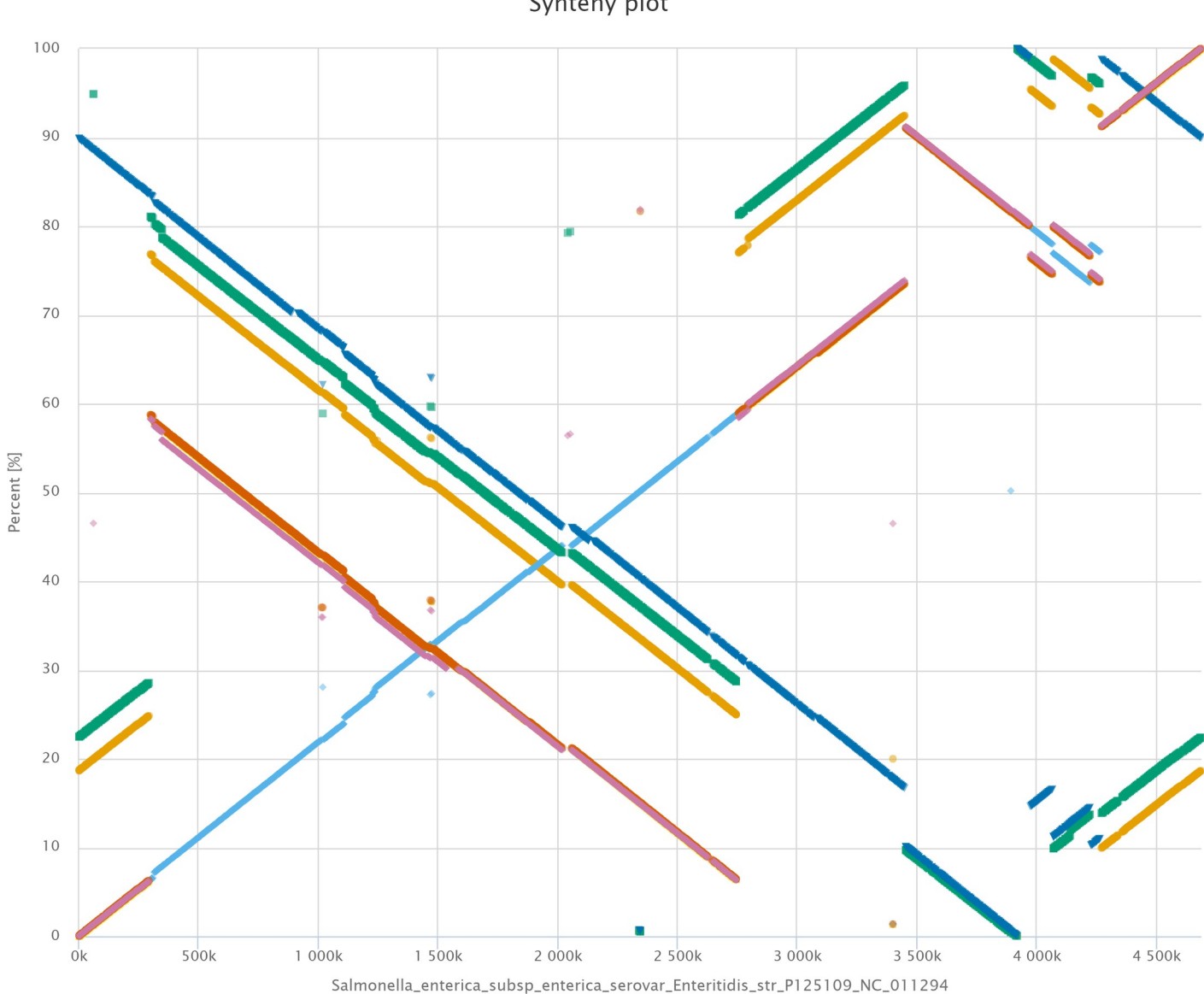

**Fig 5.** Synteny plot constructed by EDGAR of eight *S.* Gallinarum strains *i.e.* a) *S.* Pullorum str. ATCC 9120 (b) *S.* Gallinarum str. 287/91 c) *S.* Pullorum QJ-2D-Sal (d) *S.* Gallinarum/Pullorum str. CDC1983-67 (e) *S.* Gallinarum str. 9184, (f) *S.* Gallinarum/Pullorum str. RKS5078 (g) *S.* Pullorum str. S06004 against the reference of *S.* Enteritidis str. P125109.

*stf*, *stg*, *sth*, *sti*, *stj*, *stk* and *tcf* were searched for in the investigated genomes. Notably, *pef*, *sta*, *stc*, *stg*, *stj*, *stk* and *tcf* operons were not observed in any of the analyzed genomes. Manual curation of the results obtained from the tool alongwith BLASTN searches revealed that among the

fimbrial adherence determinants, chaperone-usher fimbrial gene operons of *bcf*, *fim*, *lpf*, *peg*, *saf*, *stb*, *std*, *ste*, *stf*, *sth* and *sti* were present in the *S*. Enteritidis str. P125109, however genes in the fimbrial operon *sef* were predicted to be missing in this strain (S7C Table).

On the other hand, among the investigated *S*. Gallinarum genomes, chaperone-usher fimbrial gene operons, *bcf*, *fim*, *lpf*, *peg*, *saf*, *stb*, *ste*, *stf*, *sth* and *sti* were present. Fimbrial operon *sef* and *std* operon were additionally predicted to be disrupted in the *S*. Gallinarum genomes, with *std*A and *std*B showing no significant similarity and *std*C homolog showing divergent sequence (69.11% similarity) on BLASTN analysis (S7B Table). The operon of adhesin called thin curled fimbriae (curli), which is encoded by the *agf* /*csg* fimbrial gene cluster was detected in all the 9 genomes analysed (S7C Table).

The *std* operon was detected in 3 out of 5 bvP genomes (*S*. Pullorum str. ATCC 9120, *S*. Pullorum str. S06004 and *S*. Pullorum str. QJ-2D-Sal) as well as *S*. Enteritidis str. P125109 and was absent in genomes of *S*. Gallinarum str. 287/91, *S*. Gallinarum str. 9184, *S*. Gallinarum Sal40 str. VTCCBAA614, *S*. Gallinarum/pullorum str. CDC1983-67 and *S*. Gallinarum/pullorum str. RKS5078 (S7 Table). The *sef* operon was disrupted showing complete absence of *sef*A and *sef*D genes in all the genomes. Sef operon was not detected in *S*. Gallinarum Sal40 str. VTCCBAA614 strain. The disruption in *fim* operon was only observed in *S*. Pullorum str. S06004 (S7C Table).

## 3.5 Detection of *Salmonella* Pathogenic Islands (SPIs)

A total of 113 homologs of SPIs with an average of 14.77 SPIs were detected in the investigated *Salmonella* genomes by the combined usage of web tool SPIFinder 1.0and NCBI BLAST search. The integrated approach revealed presence of high sequence identity SP-1, SP-2, SP-3, SP-4, SPI-5, SPI-12, SPI-13, SP-14, C63PI and SPCS54 islands in their genomes (S8A and S8B Table). The detected SPIs varied in the range of 0.3 to 41.8 kb in terms of size. Interestingly, 3 SPI-13 (AY956834, AY956833, AY956832) and 2 SPI-14 (AY956835, AY956836) were observed in each of the investigated genome with the exception of *S*. Gallinarum str. 9184 genome, wherein SPI-14 (AY956836) (0.4 kb) was not detected. Moreover, SPI-12 was also not observed in *S*. Gallinarum Sal40 strain VTCCBAA614. Notably, various other SPIs and resistance islands reported in *S*. *enterica* serovar such as SESS LEE, SGI-1, SPI-10, SPI-11, SPI-2, SPI-6, SPI-7, and HP1 whose sequence were extracted from PAIDB database were not found to be present by both the approaches. The features of detected SPI homologs in their respective genomes such as starting position, end position, size, and external annotation if any are detailed in (S8A and S8B Table).

## 3.6 Identification and analysis of prophage and prophage remnant regions

Each of the analyzed *Salmonella* genome was detected to possess at least one candidate prophage region by PHASTER (S2 Fig, S9 Table). In total, 23 prophage regions were identified in which fifteen were classified as putative intact regions and eight as incomplete prophage regions by PHASTER (S2 Fig, S9A Table). The detailed information of the identified prophage regions *i.e*., region name, region length, completeness, score, total protein, region position and GC percentage in the respective *Salmonella* genomes as determined by the web server are listed in S9A Table. The highest number of prophage region were detected in *S*. Pullorum QJ-2D-Sal and *S*. Pullorum str. S06004 with each possessing three complete prophages and one incomplete prophage region. In total, there were 5 different propahge elements with Gifsy_2 being the most common, which was detected in all genomes analysed. On the other hand, all 3 bvG possessed only a single (Gifsy_2) intact prophage element. Whereas bvP harboured a variety of phage elements including Gifsy_2. The average size of identified prophage regions was 36.31

kb in which largest (64.3 kb) and smallest (8.5 kb) candidate prophage regions were observed in *S*. Enteritidis str. P125109. On the other hand, the average size of identified complete pro-phage sequences in the analyzed *Salmonella* genomes was found to be 44.98 kb with lowest size of 29.2 kb belonging to *S*. Pullorum str. ATCC 9120. (S9A and S9B Table). Whereas, larg-est intact prophage region of 64.3 kb was observed in *S*. Enteritidis str. P125109 genome. The GC percentage in identified prophage region varied from 45.59% to 53.21% (S9A Table). On the other hand GC% of complete prophage regions varied from 47.23% to 53.21% (S9B Table).

### 3.7 Presence of toxin antitoxin cassettes

TA loci were identified in all the investigated *Salmonella* genomes. A total of 149 Type II TA loci were identified, ranging from 16 to 18 in each of the genome analysed (Fig 6, S10 Table). The length of the identified toxin and antitoxin proteins varied from 73–475 and 55–319 aa, respectively each. Majority of the identified candidate toxin proteins (58) harboured relElike_-domain in their structure. While, other candidate toxin proteins showed the presence of GNA-Tlike_domain (24), MazFlike_domain (9), COG2929like_domain (9), yeeU (8), doc (5), PINlike_domain (3), and NULL (24) in their structures respectively (Fig 6, S10 Table). On the other hand, identified candidate antitoxin proteins majorly possessed RHHlike_domain (60). Whilst, other domains that were observed included Xrelike_domain(27), COG5606like_do-main(9), YhfGlike_domain (9), yeeU (8), PHDlike_domain (5), doc (4), AbrBlike_domain (3) and NULL (24) in rest of the investigated candidate antitoxins (Fig 6, S10 Table). The detailed description of identified TA loci in *Salmonella* genomes which includes location in the genome, length, strand, family and domain are summarised in S10 Table.

### 3.8 Detection of acquired antibiotic resistance genes/chromosomal point mutations

Interestingly, all the analyzed *Salmonella* genomes harboured acquired aminoglycoside resis-tance gene (aac(6')-Iaa) (S3 Fig, S11 Table) as determined by Resfinder. The information related to identity, contig and position of the identified aminoglycoside gene in their respective genomes is listed in S11 Table. On the other hand, no acquired resistance gene for beta-lactam, colistin, fluoroquinolone, fosfomycin, fusidic acid, glycopeptide, MLS-macrolide, lincosamide and streptogramin B nitroimidazole, oxazolidinone, phenicol, rifampicin, sulphonamide, tet-racycline, and trimethoprim was detected in the analyzed *Salmonella* genomes by Resfinder. Strikingly, known chromosomal point mutation in *gyr*A was detected in 2 bvG and 2 bvP genomes out of the nine analysed *i.e*., *S*. Gallinarum str. 287/91, *S*. Gallinarum Sal40 strain VTCCBAA614, *S*. Pullorum str. S06004 and *S*. Pullorum QJ-2D-Sal (S11 Table). In addition, unknown mutations were detected in 16S_rrsD gene of all the analysed genomes (S11 Table). Moreover, unknown mutations in *par*E were detected by Resfinder in *S*. Gallinarum Sal40 strain VTCCBAA614 and *S*. Pullorum str. S06004. A total of six antimicrobial resistance genes (acquired, known and unknown chromosomal point mutations) were identified among the nine investigated strains.

## 4. Discussion

*Salmonella enterica* subsp. *enterica* comprises of both host-adapted and host-promiscuous pathotypes that causes a spectrum of diseases depending on the serovar or host [51, 52]. Pullo-rum Disease (PD) and Fowl Typhoid (FT) caused by *S*. Gallinarum biovars Pullorum (bvP) and Gallinarum (bvG) respectively are endemic in countries of Asia and South America lead-ing to huge economic losses to poultry industry [1, 5, 53]. It is interesting to note that as com-pared to FT, the PD reports have been low in India [54]. On the other hand, PD is frequent in

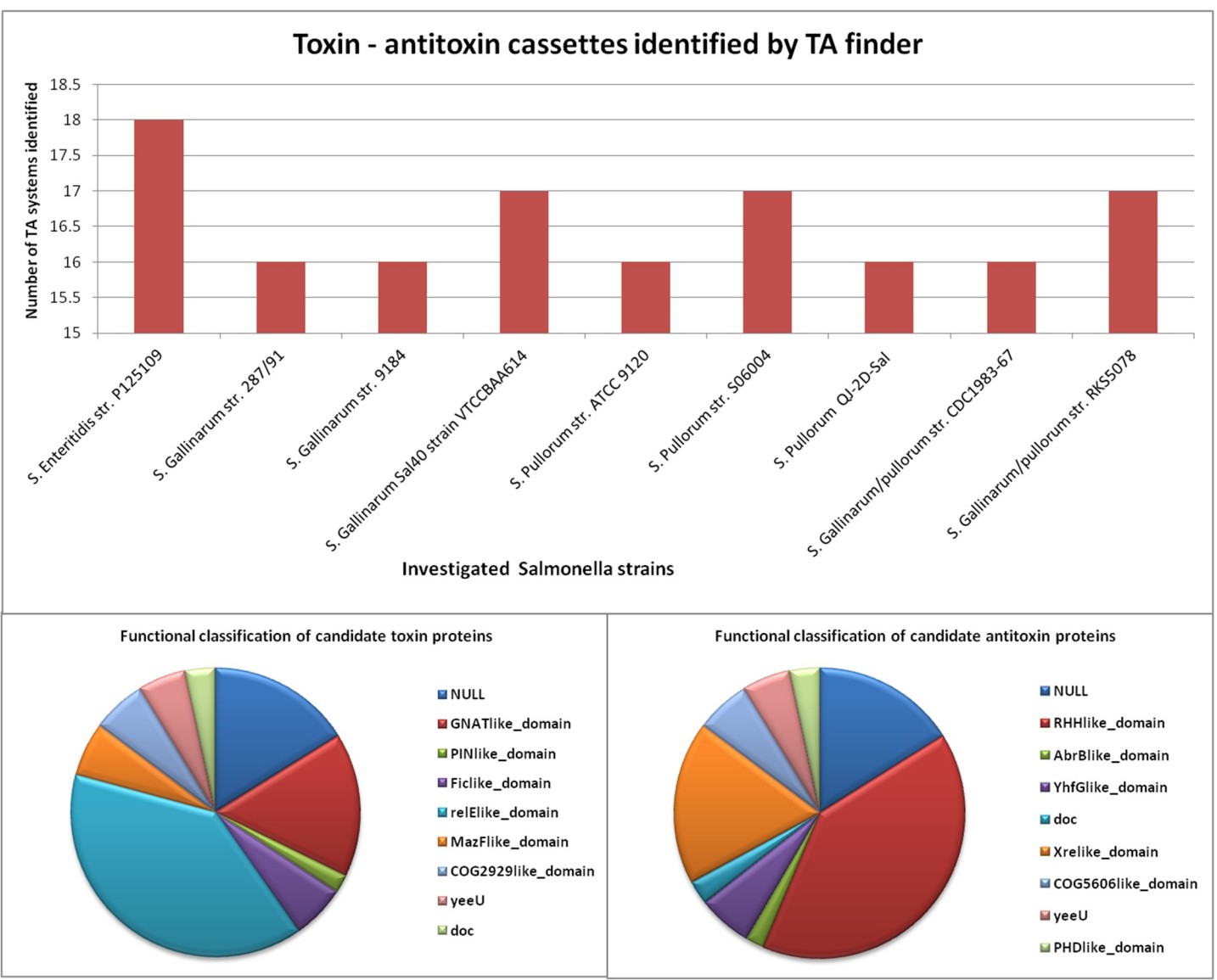

**Fig 6. Toxin-antitoxin cassettes identified by TA finder in investigated *Salmonella* strains with functional classification of candidate toxin and antitoxin proteins.**

China, being prevalent in every province [5]. Biosecurity and vaccinations applied in combination are important strategies to prevent and control these diseases in commercial and breeding flocks. Most commercially available vaccines are killed vaccines, although vaccination strategies for effective *Salmonella* control mainly include use of live attenuated strains capable of inducing a cellular immune response [55] still concerns about poor protection, lack of understanding of genetic basis of attenuation and residual pathogenicity remain [56–58].A progress can be made by a thorough understanding of the genetic virulence makeup of prevalent strains. Recently, we had obtained the first draft genome sequence of an Indian strain of *S*. Gallinarum VTCCBAA614 isolated from diseased poultry [32]. In the present investigation, we performed comparative genome analysis of eight strains of *S*. Gallinarum bvG and bvP originating from different countries (Brazil, USA, China and India) including *in house* strain with host-promiscuous *S*. Enteritidis str. P125109 taken as reference genome to characterize their

genomes and gain insights into pan genome, pathogenesis, mobiliome, resistome, and taxonomy.

For a comprehensive analysis in an open pan-genome, determination of minimum number of necessary strains is difficult [27]. The present analysis entails sufficient genotypic variability, as our selected eight genomes represent a single serovar Gallinarum of *Salmonella enterica* ssp. *enterica*, and the total SNP count in each genome varied from 7424 to 12,726 (Table 1). The pan-genome of investigated strains of *S.* Gallinarum comprises of 5091 CDS with a core-genome of 3270 CDS and an indispensible genome of 1254 CDS (Table 2). With a $R_{cp}$ of 64%, the genomes depict a fairly high degree of genomic diversity and asymmetry [59].

The pan-genome and core-genome development plot analysis in the present investigation revealed *S.* Gallinarum to possess an open pan-genome state (Fig 2A and 2B). A steady growth with addition of new genomes is depicted in the pan-genome development plot, which indicates the capacity of this sympatric species to rapidly acquire exogenous DNA [60]. The core development plot becomes limited to about 2600 genes (Fig 2B). The acquisition of exogenous DNA poses obvious challenge for control strategies against FT/PD, however core-genome based antigen selection also offers insight for vaccine candidates. It is important to examine these core genes for reverse vaccinology application as their presence in all strains and high degree of conservation makes them effective source of potentially universal antigens [61]. Although the dispensable genes are strain-restricted, they can also be explored and exploited for immune-antigens [61] (Fig 3, S6B Table). Interestingly, strain specific genes were also detected in all the investigated genomes in the range of 3–102 (S5 Table). Furthermore, singleton development plot demonstrated the possibility of finding 43 new genes with each newly sequenced genome (Fig 2C). Indian strain *S.* Gallinarum Sal40 strain VTCCBAA614 harboured the highest number of singletons *i.e.* 102 whilst *S.* Enteritidis str. P125109 possessed 96 singletons (S5 Table). Whether the presence of large number of singletons in *S.* Gallinarum Sal40 strain VTCCBAA614 and *S.* Pullorum QJ-2D-Sal, in comparison to other investigated strains is related to higher acquisition of foreign genes requires to be investigated.

Lower core-genome to pan-genome ratio and presence of large number of singletons in *S.* Enteritidis str. P125109 and other investigated strains suggests the dynamic state of *S.* Gallinarum genome. *Salmonella* Gallinarum has undergone extensive degradation and also acquired specific genes related to virulence, non-virulence, metabolism, information storage by horizontal gene transfer which may contribute to its avian host specificity [10, 27, 62].

Studies on very large number of *Salmonella enterica* genomes such as one employing 4893 genomes detected a pan-genome of 25.3 Mbp, a strict core of 1.5 Mbp present in all genomes, and a conserved core of 3.2 Mbp found in at least 96% of these genomes [63]. This makes for a strong case for creating a database of serovar specific genome components such as for *S.* Gallinarum, a host specific pathogen.

Following core-genome identification, its functional analysis by employment of eggNOG-mapper v2revealed that large number of the CDS (20%) belonged to function unknown category (5923), which indicates that our understanding of genetic repertoire of *Salmonella* genome is limited. It is probable that a sizeable number of such CDS may be mutant phenotypes for protein-coding genes [64]. Otherwise the maximum genes were related to transcription, cell-wall and membrane biogenesis, energy metabolism, amino acid metabolism and transportation among others (Fig 3, S6A Table). The identified core-genome is also dominated by ATP-binding cassette (ABC) exporters which could be explored as novel drug targets [65]. On the other hand, the functional annotation of complete set of strain specific CDS of investigated strains fell in the categories of function unknown, replication and repair, and defence mechanism among others (Fig 3, S6B Table).

The core-genome based phylogeny (Fig 4) of *S.* Gallinarum biovars. from different geographical origins has clearly divided strains into divergent clades, and Indian strain was grouped externally in the *S.* Gallinarum clade, as in previous study [1]. Similarly, the phylogenetic relationship between *S.* Gallinarum biovars and *S.* Enteritidis has been genomically analysed, which highlighted the shared ancestry of bvP and bvG with *S.* Enteritidis descendent originating in a 'second' clade [8]. In a recent study, the core-genome SNP phylogeny of only bvP strains from China has divided it into four lineages [5], which suggests that the study of within biovar genomic epidemiology of Indian bvG strains will similarly need inclusion of more number of local strains. The genomic diversity within the investigated *S.* Gallinarum strains is supported by large scale genomic rearrangement events including inversions, deletions, relocations and duplications detected in synteny analysis (Fig 5, S1 Fig).

The evolution of host-adapted serovars of salmonellae has traversed a path of periodic gene acquisition and gene disruption as a mechanism of niche adaptation away from enteric location to systemic one [8]. These genetic elements contribute to the bacterial virulence, pathogenesis, fitness in specific niche, and evolution [8, 12]. Putative virulence factors in the investigated *Salmonella* genomes related to fimbrial adherence and secretion system among others were searched and analysed (S7A–S7C Table).

The fimbrial antigens play an important role in adherence of bacteria to the cell surface, which is essential to the pathogenesis of the disease, leading to bacterial invasion [66]. Although a number of fimbriae have been described in *Salmonella*, the chaperone-usher pathway represent the major fimbriae encoded by *Salmonella* [67–72].

The degradation in fimbrial operon in *S.* Gallinarum has been associated with departure from intestinal colonization [5]. In our investigation, the fimbrial operons *Agf/Csg*, *bcf*, *lpf*, *peg*, *saf*, *stb*, *ste*, *stf*, *sth* and *sti* were observed to be present in all of the *S.* Gallinarum genomes. Suez et al, (2013) [73] performed Comparative Genomic Hybridization (CGH) on 5656 *Salmonella* ORF representing 12 invasive *Salmonella* serovars including serovar Dublin and Cholerasuis, 2 host-associated serovars, and detected five fimbrial clusters (*bcf*, *csg*, *stb*, *sth* and *sti*) to be part of core genome for invasion and systemic disease in humans. In our study, among the *S.* Gallinarum genomes, chaperone-usher fimbrial gene operons, *bcf*, *lpf*, *peg*, *saf*, *stb*, *ste*, *stf*, *sth* and *sti* were detected to be part of the core genome. Notably, in the *S.* Gallinarum genomes studied, operon *fim*, *sef*, and *std* were predicted to be disrupted (S7C Table).

Hu et al, (2019) [5], reported *std* along with *saf* and *csg* to be intact in almost all the *S.* Pullorum strains in their study. On the other hand, we detected differential distribution of *std* operon in our study. The *std* operon, which was detected in *S.* Enteritidis str. P125109 was not detected in any of the 3 bvG, and 2 of the 5 bvP genomes studied (i.e., *S.* Gallinarum/pullorum str. CDC1983-67 and S. Gallinarum/pullorum str. RKS5078). In addition to *std* operon, the *sef* operon, *sef*A and D genes were not detected in *S.* Enteritidis str. P125109 and all other investigated genomes.SEF14 plays a role in the colonization of Peyer's patches and in the adhesion and invasion of intestine epithelial cells [74–76].

The *lpf*, *fim*, and *agf /csg* operons, which were detected in all of our investigated genomes, encode fimbrial proteins which mediate attachment of *S. enterica* serotype Typhimurium in epithelial cell lines *in vitro* for long-term intestinal persistence [77]. The gene *csg*D of *agf /*csg operon activates the production of curli fimbriae by transcriptional activation of the *csg*BAC operon encoding the structural genes of curli fimbriae [78]. The *csg*A gene of *agf /csg*operon has also been evidenced to be involved in biofilm formation in *S.* Pullorum, which may contribute to the virulence [79].

The type 1 fimbriae (T1F) are important virulence factors in *Enterobacericeae* including *Salmonella* spp., and is commonly expressed in virulent strains of *Salmonella* spp. [80, 81]. In our study, the *fim* operon was detected in all genomes except absence of *fim*A and I genes in *S.*

Pullorum str. S06004 genome. The genes responsible for synthesis of T1F are *fim*I, *fim*C, *fim*D, *fim*H, and *fim*F, and *fim*A promoter region making a single operon [82].

In all the genomes studied, *peg*ABCD operon was detected to be the part of core genome. The peg fimbrial gene cluster originally discovered in *S. enterica* subsp. *enterica* serovars Paratyphi A, Enteritidis, and Gallinarum [10], is considered to influence caecal colonisation of chickens by *S.* Enteritidis [83]. However, Hu et al, 2019 [5], found the peg operon, disrupted in all strains of *S.* Pullorum, suggesting their redundancy.

The Salmonella pathogenicity island 1 (SPI-1), encoding type III secretion system (T3SS) secreted effector genes, consisting of *sop*A, *sop*B, *sop*D, *sop*D2, *sop*E and *sop*E2 are involved in the inflammation and diahhorea [84]. *sop*A, *sop*B, *sop*D, *sop*E and *sop*E2 were detected in all genomes except in *S.* Gallinarum strain 9184. Among the Salmonella pathogenicity island II (SPI-II) encoding T3SS effectors (*pip*B2, *sse*G, *sse*I, *ssek*2, and *ssa*S), the *sse*I and *ssek*2 were detected in *S.* Enteritidis str. P125109, however, *sse*I was not detected in any bvG and bvP strains, and *sse*K2 was absent in 3 bvG (str. 287/91, str.9184, str. VTCCBAA614) and 1 bvP strain (str. CDC1983-67). The SPI-2 effectors (*sse*I, *sif*A, *sse*F, *pip*B2, *spi*C, *ssp*H2, and *slr*P) have been shown to play a role in inflammation, particularly in inhibition of dendritic cell migration from intestinal environment [85]. Recently, it has been demonstrated that pseudogenization of *sse*I in sequence type 313 (ST313) *Salmonella* Typhimurium has rendered it invasive in sub-Saharan African region human subjects [86]. Our BLAST results clearly show the genes to be truncated in all bvG/bvP strains studied (S7B Table). *sse*I is a also a pseudogene in *S.* Enteritidis isolates which cause systemic infection within immunocompromised hosts [87].

We next focussed our investigation on identification and analysis of candidate virulence factors that were previously not identified in the genomes of the analyzed strains which included genomic islands, prophages, TA cassettes and AMR genes.

The acquisition of virulence capability enhancement *en masse* by acquisition of SPI has been hall mark of *S.* Gallinarum [17, 88]. A total of 113 SPI homologs with nearly 15 SPIs per genome were detected by combinatorial usage of SPIfinder and BLAST searching (S8A and S8B Table), which uncovered the presence of homologs of SP-1, SP-2, SP-3, SP-4, SPI-5, SPI-12, SPI-13, SP-14, C63PI and SPCS54 island in the investigated genomes (S8A and S8B Table). The presence of particular SPIs in each of the genomes indicates their specificity towards *S.* Gallinarum and *S.* Enteritidis and thus corroborates with previously published reports of SPIs showing serovar specificity [44, 89, 90].Notably, the role of SPI's other than SP1 and SP2 in virulence and pathology of *S.* Gallinarum infections has not been analyzed in detail. The elucidation of functional role of other SPIs thus requires further experimental investigations with functional knockout mutations or via targeting and deletion of SPIs by utilization of CRISPR-Cas systems [91–94]. The limitation of manually screening mutant phenotypes in functional knockout mutation studies has led to use of high-thoroughput sequencing technologies in conjunction with transposon mutagenesis i.e., transposon sequencing (Tn-seq) [95, 96]. Transposon-directed insertion site sequencing (TraDIS) was used to study the mechanisms used by *Salmonella* to enter and persist within the bovine lymphatic system in comparison to intestinal colonization [97]. Several variations of Tn-seq have been devised which mainly differ in transposon junction sequence amplification techniques, which can be applied to obtain global gene functional information in *Salmonella*.

Notably, SPI-12 (NC_006905_P4, 11.1 kb), and SPI-14 (AY956836, 0.4kb) were not detected in *S.* Gallinarum Sal40 strain VTCCBAA614 and *S.* Gallinarum str. 9184, respectively. SPI-12 has been shown to be crucial for bacterial survival in the host [98]. SPI-14 has been recently identified as being involved in *Salmonella* intestinal survival and invasion via activation of SPI-1 genes [99].

Prophages contribute to pathogenicity, bacterial fitness, diversity, resistance to phages, antimicrobial resistance, evolution as well as aid in increased environmental tolerance [20, 100, 101].

In our study, the intact prophages were detected in the range of 1–3 with the highest number in bvP QJ-2D-Sal and bvP str. S06004. As in our study, a recent study also detected only a single phage in all of their bvG genomes, which is less than 5±3 prophage regions per genome detected in 1,760 *S. enterica* genomes, using Phaster [102]. However the observed prophage diversity in our study is in contrast with lower diversity reported by [103], using 2 strains of *S*. Gallinarum. The gifsy_2 prophage element which was detected in all genomes analysed in this study was earlier detected predominantly in *S*. Typhimurium genome [102]. Significantly prophage element ST104 detected in bvP QJ-2D-Sal has been previously reported in MDR *S*. Typhimurium DT104 as phage ST104. Notably, prophage and plasmids acquisition has been demonstrated in *S*. Pullorum leading to emergence of multi-drug resistant (MDR) strains [5, 104]. Our study also highlights variable number of propahges detected in bvP as compared to single prophage element detected in bvG. Hu et al, 2019 [5] also detected multiple prophage elements in their bvP genomes, which leads to high genetic diversity besides AMR. The differential presence of prophages in the current study indicates the ability of *S*. Gallinarum genomes to acquire new gene content, and introduce diversity in them which is in accordance with the singleton development plot that depicted the finding of 43 new genes with each newly sequenced genome. This is further refurbished by detection of holin, phage-like protein, phage membrane protein, side tail fibre protein genes among 104 singletons detected in *S*. Gallinarum Sal40 strain by EDGAR (S4I Table)). The differential distribution of prophages in the current study can be utilized as markers for strain characterization, diversity assessment, tracking of strains, determination of transmission history of *S. enterica* Gallinarum strains as phage typing has been widely employed by various investigators for these goals in various bacterial species [21, 105–109]. Moreover, cryptic prophages identified in the study can be utilized as drug target after determination of their role and essentiality status in the genome as they are permanent resident of microbial genomes [110].

In order to elucidate the role of prophages of *S*. Gallinarum, prophage induction may be achieved using the SOS response activated by chemicals such as mitomycin C, hydrogen peroxide and UV radiation, which cause DNA damage, apart from instances of spontaneous induction, and inflammation [111–113]. Induced prophages can be further studied by genomic sequencing and bioinformatic methods [112]. Further, specific prophage methylation pattern and repertoire of methyltransferase motif estimation can give useful insights on prophage gene expression [114]. Nothwithstanding the widespread repression of prophage genes, transcriptomics data analysis of propahge regions can be an important way forward [112, 115].

Toxin-antitoxin operons which are implicated in pathogenicity, virulence, persistence, stress endurance, and antibiotic resistance in bacteria and archae were searched for in the investigated *Salmonella* genomes [116–118]. The computational search revealed a total of 149 TA cassettes in the range of 16–18 in the investigated *S*. Gallinarum genomes (S10 Table). Significantly, a majority of the toxins of identified TA systems belonged to relElike family and highest number of identified antitoxins possessed RHH like domain (S10 Table, Fig 6). The presence of 149 Toxin antitoxin operons in the range of 16–18 in each of the analyzed *S*. Gallinarum genomes indicates their possible role in growth regulation, niche adaptation, encountering of various stress responses inside macrophage as well as in persistence, as has been observed in various bacteria and archaea including *S*. Typhimurium LT2, [24, 117, 119]. Moreover, the biological significance of presence of large number of TA's may be due to their potential involvement in number of underlying regulatory pathways, as can be inferred by identification of different domains such as GNATlike_domain (24 nos.), MazFlike_domain (9 nos.), COG2929 like_domain (9 nos.), yeeU (8 nos.), doc (5 nos.), and PINlike_domain (3 nos.) in the structure of the toxins detected in the study. These TA domains have been previously reported to be associated with distinct features and different functional roles [117, 120]. As the acquisition of TA operons can be either through horizontal gene transfer or gene

amplification [121], therefore the diversity of TA operons observed in *Salmonella* genomes reemphasizes the flexible nature of *S*. Gallinarum genome. Elucidation of their role requires further experimental analysis to be carried out.

The *in silico* identified TA systems also requires to be experimentally characterized before considering them as bonafide TA operons by designing assays to test whether toxicity of over-expressed toxin protein of TA pair is abrogated by over-expression of putative antitoxin protein in heterologous systems [122–124]. Moreover, experimental investigation is required to decode their role in pathogenesis. In addition, to knock out studies a number of powerful genetic tools can be employed to elucidate the function of detected virulence factors. Ectopic expression of TA systems can be studied to decipher their role in growth regulation [125, 126]. In addition, site directed mutagenesis can be utilized for detection of residues crucial for toxin activity for functional characterization of TA systems [127, 128]. Moreover, expression profile of identified TA modules in response to various stresses at multiple time points can be detected by using whole genome micro-array platform comprising of TA loci probes and that could lead to co-expressed genes/pathways [129]. The *Salmonella* TA systems can be considered as potent targets for structure based inhibitor design as has been previously employed against various other bacteria [130, 131].

Although plasmids are important carriers of AMR genes in *Salmonella* serovars, [132], we detected the presence of acquired aminoglycoside resistance gene (aac(6')-Iaa) in all the investigated *Salmonella* genomes (S3 Fig, S11A Table). Neuert *et al*., 2018 [133] found in their study on Non-typhoidal *Salmonella* (NTS), that nearly all but eight of the total 3,491 isolates carried an aminoglycoside acetyltransferase *aac*(6')-type gene. However, only eleven showed phenotypic resistance to an aminoglycoside antimicrobial. In another study, the genes associated with *ant* and *aph* were found in all aminoglycoside-resistant *Salmonella* isolates, but *aac* genes were only found in *Salmonella* ser. Typhimurium and *Salmonella* ser. Heidelberg from chicken-sourced isolates [134]. The observation is in concordance with previously published reports stating that aminoglycosides exhibit weak bactericidal activity against intracellular *S. enterica* serovars [135]. Strikingly, known mutation in *gyr*A were also observed in four genomes out of the nine investigated i.e., *S*. Gallinarum str. 287/91, *S*. Gallinarum Sal40 strain VTCCBAA614, *S*. Pullorum str. S06004 and *S*. Pullorum QJ-2D-Sal (S11B Table). Koerich et al (2018) [58] also reported high resistance to drugs from macrolide and quinolone groups in *Salmonella* Gallinarum field isolates. Literature mining has revealed that mutations in *gyr*A are associated with resistance to fluoroquinolones in eight species of *Enterobacteriaceae* [136]. Chromosonal point mutations in *rrs*D gene in the helix 34 region, which results in changes to the drug-binding site in the ribosomal 16S rRNA has been reported to confer resistance against aminoglycoside spectinomycin of *Salmonella enterica* serovar Typhimurium gene [137, 138]. Workers have also reported mutations in *rrs*D genes in 2 *S. enterica* isolates from cattle and chicken in South Africa [139].

Antimicrobial resistance in *Salmonella* serovars is a serious poultry husbandry and public health problem all over the world including Asian, and South American region [58, 140, 141]. The observation of variable Antimicrobial resistant genes in the study can be utilized as AMR markers. The presence of additional AMR genes in *S. enterica* Gallinarum str. Sal40 and *S. enterica* Pullorum str. S06004 indicates the ability of acquiring exogenous DNA for adaptation in response to environmental requirements. The differential presence underlines the need of NGS based mapping of AMR genes in *S*. Gallinarum serovars to better manage their control measures [142].

A number of investigators have previously carried out comparative genome analysis of *S*. Gallinarum with several other serovars which includes *S*. Enteritidis, *S*. Typhimurium to decode the genomic differences amongst them. The comparative genome analysis of *S*. Enteritidis PT4, *S*. Typhimurium LT2 and *S*. Gallinarum 287/91 was carried by Thomson *et al*., (2008) [10] wherein they reported predominant similarity and synteny in their core genomes and overrepresentation of putative pseudogenes in *S*. Gallinarum 287/91 in comparison to *S*.

Enteritidis str. P125109. Similarly, a genomic comparison between *S*. Gallinarum and *S*. Pullorum was carried out by Feng and associates [143] and they included *S*. Gallinarum str. CDC1983-67, *S*. Gallinarum str. RKS5078, *S*. Gallinarum str. 287/91, and *S*. Enteritidis str. P125109 and reported a high number and differential distribution of pseudogenes in bvG and bvP strains in reference to *S*. Enteritidis strain. On the other hand, genome comparison between closely related *S. enterica* serovars Enteritidis, Dublin, and Gallinarum revealed differential distribution of prophages and pseudogenes in their genomes along besides SNPs, insertions, and deletions amongst the investigated strains [103]. In a similar manner, to decode different host-specificities, genes related to SPI of *S*. Enteritidis PT4 (NCTC 13349) and *S*. Gallinarum 287/91 NCTC 13346 were compared by Eswarappa et al. 2009 [144] and detected 24 positively selected genes that included SPI-2 TTSS and effector proteins of SPI-1 TTSS.

The present study suggests that the host restricted *S*. Gallinarum strains harbour strain specific genes and they exhibit differential distribution of putative virulence factors such as genomic islands, prophage regions, TA cassettes, and acquired AMR genes in their genomes. This study also highlights the need to analyse more number of *S*. Gallinarum genomes to understand the phylogeny and better capture of its pan-genome biodiversity. This comparative genomic analysis of *S*. Gallinarum serovar has provided a valuable insight and laid foundation for future experimental studies to be carried out to decipher the underlying mechanisms driving the pathogenesis and virulence of this avian restricted pathogen.

## 5. Conclusion

*Salmonella enterica* serovar Gallinarum biovar Pullorum (bvP) and biovar Gallinarum (bvG), are the causative agents of disease (PD) and fowl typhoid (FT) respectively, which causes considerable economic losses to poultry industry worldwide especially in developing countries including India. Comprehensive comparative genome analysis of eight *S. enterica* serovar Gallinarum strains originating from different geographical regions including Indian strain *S*. Gallinarum Sal40 VTCCBAA614 was carried out to decode the genotypic differences amongst them with a focus on detection and analysis of candidate virulence factors. The investigation revealed an open pan-genome for *S*. Gallinarum encompassing 5091 coding sequence (CDS) with 3270 CDS belonging to core-genome, 1254 CDS to dispensable genome and strain specific genes amongst the analyzed strains. Furthermore, analysis of distribution of candidate virulence factors in the investigated strains revealed diversity and differential distribution of genomic features such as genomic islands, prophage regions, toxin-antitoxin operons, and acquired antimicrobial resistance genes. The fimbrial operons *Agf/Csg*, *bcf*, *lpf*, *peg*, *saf*, *stb*, *ste*, *stf*, *sth* and *sti* were observed to be present in all of the *S*. Gallinarum genomes, whereas operon *sef*, and *std* showed differential distribution. The computational search unravelled the existence of high sequence identity genomic islands SP-1, SP-2, SP-3, SP-4, SPI-5, SPI-12, SPI-13, SP-14, C63PI and SPCS54 in their genomes. Additionally, 23 prophage regions and 149 Type II TA loci were also identified and characterized in the investigated genomes. The genomic variability detected among the *S. enterica* serovar Gallinarum strains will form the basis for future experimental investigations and could be used for bacterial typing. The core genome assignment can be utilised towards design of diagnostics, as well as drug and vaccine target prediction for effective surveillance, prevention, and control.

## Supporting information

**S1 Fig.** (a-h) Synteny plot analysis of investigated *Salmonella* strains with reference to *S*. Enteritidis str. P125109. Depicts synteny plot analysis of investigated *Salmonella* strains. (TIFF)

**S2 Fig. Prophages detected by Phaster in analyzed *Salmonella* strains.** Details of prophages detected.
(TIF)

**S3 Fig. Acquired antimicrobial resistance genes, known chromosomal mutations, and unknown chromosomal mutations detected in investigated *Salmonella* strains by Resfinder.** Detects acquired antimicrobial resistance genes.
(TIF)

**S1 Table. Pan-genome and core genome development projections for investigated nine *Salmonella* strains.** Pan-genome and core genome development projections.
(DOCX)

**S2 Table. Various features of investigated genomes such as sequencing technology, genome coverage, assembly method and genome quality measures.** Quality features of analyzed genomes.
(XLSX)

**S3 Table. List of CDS that are part of pan genome of investigated nine *Salmonella* strains with their name and function.** Pan genome CDS list of genomes.
(XLSX)

**S4 Table. List of CDS that are identified to be part of the core genome of analyzed nine *Salmonella* strains.** Core genome CDS list of genomes.
(XLSX)

**S5 Table.** (a to i) Description of singletons detected in investigated *Salmonella* Gallinarum (a) *S.* Enteritidis str. P125109 (b) S. Gallinarum/pullorum str. CDC1983-67 (c) S. Gallinarum/pullorum str. RKS5078 (d) *S.* Gallinarum str. 287/91 (e) *S.* Gallinarum str. 9184 (f) *S.* Pullorum str. ATCC 9120 (g) *S.* Pullorum str. S06004 (h) *S.* Pullorum QJ-2D-Sal (i) *S.* Gallinarum Sal40 strains by EDGAR with their name and function with reference to *S.* Enteritidis str. P125109. Detected Singletons listing of all genomes.
(XLSX)

**S6 Table.** (a-b) The functional annotation of (a) Suppl Table 6 (a): This data sheet consists of functional annotation of core genome of analyzed Salmonella strains performed by eggNOG-mapper v2 and lists query, seed ortholog, e-value, score, best taxonomy level, preferred name, GO terms, EC number, annotation level, COG category and description. (b) Suppl Table 6 (b): This data sheet consists of functional annotation of strain specific coding sequences of analyzed Salmonella strains performed by eggNOG-mapper v2. Functional annotation of CDS of analysed genomes.
(XLSX)

**S7 Table.** a. The table list virulence factor (VF) identified by VFDB database in the investigated Salmonella genomes. b. Details of virulence factor (VF) identified by BLASTN searches showing differential distribution in VFDB results in the investigated Salmonella genomes. c. Consolidated details of virulence factor (VF) identified by VF analyzer tool and BLASTN searches in the investigated Salmonella genomes. Virulence genes.
(XLSX)

**S8 Table.** (a-b) The datasheet lists homologs of *Salmonella* pathogenic islands (SPIs) detected by (a) SPIFinder 1.0 and (b) BLAST search, in the investigated *Salmonella* genomes. Listing of

Salmonella pathogenicity island detected by SPI Finder and by BLAST search.
(XLSX)

**S9 Table.** (a-b) The identified (a) complete list of prophage regions and (b) intact prophages, in the genomes of nine *Salmonella* strains that were investigated in the study with description of a) region number b) region length c) completeness d) score e) total proteins f) region position g) most common phage and h) GC%. The colour coding of the rows represents the classification of identified prophage regions by PHASTER, green- intact prophage region, red- incomplete prophage region and blue- questionable prophage region. Description of prophages detected in all genomes with details.
(XLSX)

**S10 Table. This worksheet contains 149 TA gene cassettes identified in nine *Salmonella* genomes by TA finder.** TA gene listing detected in all genomes.
(XLSX)

**S11 Table.** (a-c) This datasheet details (a) acquired antimicrobial resistance genes, (b) known chromosomal mutations, and (c) unknown chromosomal mutations detected in investigated *Salmonella* strains by Resfinder. Details of AMR genes detected in genomes.
(XLSX)

## Author Contributions

**Conceptualization:** Rajesh Kumar Vaid.

**Data curation:** Zoozeal Thakur.

**Formal analysis:** Zoozeal Thakur.

**Funding acquisition:** Rajesh Kumar Vaid, Sanjay Kumar.

**Investigation:** Taruna Anand.

**Methodology:** Rajesh Kumar Vaid, Zoozeal Thakur.

**Project administration:** Rajesh Kumar Vaid.

**Resources:** Rajesh Kumar Vaid, Bhupendra Nath Tripathi.

**Software:** Sanjay Kumar.

**Supervision:** Rajesh Kumar Vaid, Bhupendra Nath Tripathi.

**Visualization:** Taruna Anand.

**Writing – original draft:** Zoozeal Thakur.

**Writing – review & editing:** Rajesh Kumar Vaid, Taruna Anand, Sanjay Kumar, Bhupendra Nath Tripathi.

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
