## [Decision Letter · Decision Letter 0]

17 Mar 2021

PONE-D-21-04830

Comparative genome analysis of Salmonella enterica serovar Gallinarum biovars Pullorum and Gallinarum decodes strain specific genes

PLOS ONE

Dear Dr. Vaid,

Thank you for submitting your manuscript to PLOS ONE. After careful consideration, we feel that it has merit but does not fully meet PLOS ONE’s publication criteria as it currently stands. Therefore, we invite you to submit a revised version of the manuscript that addresses the points raised during the review process.

Both reviewers enjoyed reading the work conducted, but both had some pointed comments that will require some major editorial alterations tot eh manuscript.  Reviewer #1 would lie to see an expansion of the discussion on putative virulence genes identified. specifically a better idea of what future research will be necessary to elucidate the function of these genes.  Further, this reviewer doubts the validity of the idea of creating mutants for thiese virulence genes because of the large volume of work already published on virulence of Salmonella and whether any novel findings will be produced.  Reviewer #2 had many more comments that require close attention.  This reviewer had 3 major concerns: 1. the adequacy of the sample size (n=8).; 2. the descriptive nature of the study with no clear goal/hypothesis; and 3. the approach of comparative genomics  used in these experiments.  Your revised manuscript will require detailed answers or rebuttals to all three concerns by this reviewer..

We look forward to receiving your revised manuscript.

Kind regards,

Michael H. Kogut, Ph.D.

Academic Editor

PLOS ONE

Journal Requirements:

3. Please upload a copy of Figure 7, to which you refer in your text on page 26. If the figure is no longer to be included as part of the submission please remove all reference to it within the text.

4. Please include your tables as part of your main manuscript and remove the individual files. Please note that supplementary tables should remain as separate "supporting information" files

Reviewers' comments:

Reviewer's Responses to Questions

**Comments to the Author**

1. Is the manuscript technically sound, and do the data support the conclusions?

Reviewer #1: Yes

Reviewer #2: Partly

2. Has the statistical analysis been performed appropriately and rigorously? 

Reviewer #1: N/A

Reviewer #2: No

3. Have the authors made all data underlying the findings in their manuscript fully available?

Reviewer #1: Yes

Reviewer #2: Yes

4. Is the manuscript presented in an intelligible fashion and written in standard English?

Reviewer #1: Yes

Reviewer #2: Yes

5. Review Comments to the Author

Reviewer #1: This manuscript describes the comparative genomic analysis of a new strain of Gallinarim from India in the context of previously sequenced and analyzed strains. The goal of the investigation was to determine the pangenome, the accessory genome, and strain specific gene176, s that would hopefully help them identify virulence genes. The authors did a good job of all but the last of these goals. They did identify a few SPIs that seem to be unique to the Gallinarim and Poullorum serotypes, and a few of these have been associated with virulence in the past. However, they ended up with a long list of genes, phage, toxin-antitoxin systems, etc. that have no known function. While these can be targeted in future research, these list of genes with unknown functions seem to be the results of many comparative genomics studies that hope to find the virulence genes or other genes that explain pathogenicity of serotypes or their host range. Some specific points the authors may consider to address follow.

1 Please expand the discussion on all of these putative virulence genes so the reader has a better idea of what future research will be necessary to elucidate the function of these genes.

2 While suggesting doing knockout mutants and complementation to discover the function of these genes, it seems unlikely this will yield results and Salmonella has been the target of intensive mutant screening for virulence factors. It is more likely that many of these systems in Salmonella are redundant and mutants show very little or nonsignificant phenotypes. What other approaches could be done?

3. Line 126 rather than enlisted consider listed.

4. Line 200 why is bmrB listed twice?

5. Line 205 consider 4,657,781bp rather than 46,57,781bp

6. line 206 ditto above

7. Figure 4, what level of confidence is this tree? How many bootstraps were done?

8. Line 336, listed vs. enlisted

9532-536, this aminglycoside resistance gene is almost every Salmonella and it is non-functional due to a promotor mutation. It can occasionally mutate or have a promotor inserted in front of it and become active, but that is very rare.

Reviewer #2: Vaid et al., conducted comparative genomics of a few (n=7) published Salmonella Gallinarum (host-restricted Salmonella serotype), S. Pullorum (another host-restricted serotype), and S. Enteritidis (a broad host range serotype) along with one strain of S. Gallinarum isolated from India and sequenced in this study. The authors then present the inventory of genetic differences and describe the results of the comparative genomic analysis of this small subset of host-adapted Salmonella serovars Gallinarum and Pullorum strains in this study. The manuscript is well written, and the analysis is appropriate. In general, it is definitely of interest to study these genomes from poultry adapted Salmonella and tease out the differences to address interesting questions related to mechanisms underlying host-adaptation, bacterial evolution, identifying markers for molecular detection, differentiation, or developing strategies for vaccine and or other control measures. However, it is unclear what was the goal of this study. Second, I also wondered what is new/novel finding in this study beyond what we already know about the genetic makeup of S. Gallinarum and Pullorum from multiple studies that have already been conducted on comparing genomes to demonstrate differential gene content (eg., Thompson et al., 2008; Fricke et al., 2011; Feng et al., 2013; Liu et al., 2002; Wu et al., 2005; Tang et al., 2013; Batista et al., 2018, etc)? My major concern is that the study, as presented is descriptive and very limited in scope because it includes a minuscule sample size (n=8). In comparative genomics studies, the associations and inferences based on small sample size quickly fail (and become irrelevant) as soon as more data becomes available. A quick survey of the NCBI GenBank database shows that the data for there are 100s of Gallinarum and Pullorum sequenced strains exist (eg., https://www.ncbi.nlm.nih.gov/pathogens/isolates/#Gallinarum). Given this information, it makes more sense to include the sequences of all existing strains (from different geographic areas) in the comparative genomic analysis for reliably inferring pan-genomes and identifying serovar specific gene content, to address the objective of the study, as stated. Another concern is the use of the S. Enteritidis genome as a reference here. If you want to comparative genomics of S. Gallinarum, use a reference S. Gallinarum genome and include other Salmonella such as S. Enteritidis as control/outgroup. Another approach is to use non-reference-based comparative genomics to tease out the differences using large sample sizes of Gallinarum/Pullorum strains and then scanning for serovar specificity via broader comparative genomics analysis with Salmonella enterica genomes. Unless the sample size is increased to include all available Gallinarum and Pullorum sequences in the comparative genomics analysis, the current analysis with a few strains is extremely limited and raises questions about the validity of the inferences made here, which dampens my enthusiasm for this work. I suggest that authors provide a clear hypothesis and goal of their study and focus the analysis and discussion around those goals rather than describing every difference that is detected. In other words, differences exist, however, what’s important is what is the biological significance of these differences? For instance, S. Gallinarum and S. Pullorum have evolved over the years with extensive genomic degradation leading to the host adaptation [Fricke et al. (2011), Seif et al., (2018), Batista et al., (2018), etc], however, this has not been discussed at all in this paper other than briefly mentioning it in the discussion (line 440). It is important to recognize that the host adaptation is what makes S. Gallinarum so different from the rest of the Salmonella serovars. The host adaptation is complex and involves multiple factors with a significant implication on metabolic adaptation in a particular host. So simple inventory of genetic differences doesn’t necessarily mean much unless the significance of such changes is discussed with the relevant biological phenomenon.

6. PLOS authors have the option to publish the peer review history of their article (what does this mean?). If published, this will include your full peer review and any attached files.

Reviewer #1: No

Reviewer #2: No

---

## [Author Response · Author response to Decision Letter 0]

16 Jul 2021

Response to Reviewers

Dear Reviewer #1: 

We are indeed delighted and indebted to you for your critical reading insightful comments and suggestions for improvement of the manuscript. The Corona outbreak delayed our revision; however, we have now incorporated all of your suggestions into the revision in our manuscript, which were very helpful. 

We agree that expansion on the virulence genes elucidated will be conducive to future work. In fact your suggestion prompted us to re-analyse the VFanalyzer tool data (S7 Table), so that all genes showing differential distribution are verified, before we discuss these, and therefore, these virulence genes were further evaluated by BLASTn, and tabulated (S7 b, c Table). Similarly, your point on knockout mutant is also well taken, thereby improving our manuscript, as we have added discussion on novel approaches of functional genomics. This has been done on all sections of virulence genes i.e., virulence genes, TA genes, and prophages. Regarding the tree query, we e-mailed the EDGAR Software staff and after receiving their reply, we have mentioned pertinent points in Material & methods and results. We have also incorporated the technical data (SH branch support values), thus improving the tree figure also (See Fig 4). The point you have mentioned about aminoglycoside is very pertinent, and we have discussed its non-functionality. We are also thankful to you for bringing out other small nagging mistakes. 

We have given a detailed query wise response below as per your comments.

Thank you for your time and kind help.

regards,

1. Query: 

Please expand the discussion on all of these putative virulence genes so the reader has a better idea of what future research will be necessary to elucidate the function of these genes.

1. Response:

We find merit in the observation of learned reviewer and have taken up discussion of the virulence genes in a more objective manner, specifically targeting Suppl Table 7 listing virulence factors discovered. We have, in addition added to analysis work by employing BLASTn tool for checking differentially detected virulence gene sets of Suppl Table 7a, and have prepared added Table 7b and 7c, and discussed the findings. We have then discussed the tabulated virulence factors on aspects of their differential distribution in Line 521-572). 

We have further discussed the function elucidation techniques (for SPI, see Line 601-614), prophages (653- 662); TA (691-698). Apart from this, the application aspects and biological significance has been discussed (See, for Prophages Line 621-623) and 638 to 651); For TA (Line 669-685); and AMR (Line725-733). (The aspects of future research have been additionally mentioned in response to 2nd query below) 

2. Query:

 While suggesting doing knockout mutants and complementation to discover the function of these genes, it seems unlikely this will yield results and Salmonella has been the target of intensive mutant screening for virulence factors. It is more likely that many of these systems in Salmonella are redundant and mutants show very little or non-significant phenotypes. What other approaches could be done?

2. Response

We have included other methods and studies in discussion which can be employed for functional characterization of genes, apart from knockout studies previously mentioned. For studying TA systems, we have discussed ectopic expression, site directed mutagenesis, and whole genome microarray. For functional elucidation of prophages, viz., prophage induction and pathogenesis studies, genomic sequence analysis, DNA methylation pattern and location, and transcriptome data mining can be utilised. The elucidation of functional role of SPIs by functional knockout mutations and CRISPR-Cas systems has been mentioned. In addition we have also mentioned the application of Tn-sequencing to as a functional genomics tool.

3 Query:

 Line 123 rather than enlisted consider listed

3 Response:

 Correction done as suggested by reviewer

4. Query:

 Line 208 why is pmrB listed twice

4. Response:

Correction of typo done as suggested by reviewer 

5 Query:

Line 213 consider 4,657,781 bp rather than 46,57,781 bp 

5 Response:

Yes, done as suggested by reviewer

6. Query:

 line 214 ditto above 6 Response:

 Correction done as suggested by reviewer

7 Query:

What level of confidence is this tree? How many bootstraps were done?

7: Response

The FastTree was software used to construct the tree. Program employs Shimodaira-Hasegawa (SH) branch support values instead of bootstrapping to verify the tree topology. SH support values for our tree were very good in general, with a minimum value of 0.772 and only two values below the maximum of 1.00. SH values below one were added to the tree image, and a description has been added to the figure legends. 

8: Query Line 336, listed vs. enlisted

8: Response Used listed as righty suggested by reviewer (Line 384)

9 Query:

532-536, this aminoglycoside resistance gene is (in) almost every Salmonella and it is non-functional due to a promotor mutation. It can occasionally mutate or have a promotor inserted in front of it and become active, but that is very rare.

9 Response: 

This factual comment by learned Reviewer is noted and changes have been brought out in discussion text to contextualize this information (see Line 704-710) and 719 to 724.

Dear Reviewer #2: 

We are indeed indebted to you for such a deep and critical reading, and exhaustive, insightful comments and suggestions for improvement of the manuscript. We are delighted by your appreciation of manuscript writing and our analysis. Due to Corona outbreak affecting my colleague, the revision got delayed; however, we have now tried to incorporate your suggestions into the revision in our manuscript, which were very helpful. 

We appreciate your concern about goal of study and have put forth our perspective on it. We feel that you will understand the need for incorporation of descriptive analysis of S. Gallinarum strain from Indian subcontinent, which was hitherto lacking, apart from our study on S. Gallinarum serovars. In order to mention, that what clearly demarcates our study from the studies (References) which you have quoted, we have specially prepared a Response Table 1 clearly bringing about the differences of these studies from our work. 

Like your kind self, we were also seized with your concern about sample size (and that is why it was even part of discussion of our paper); however you would appreciate the limitation on availability of full length serovars, which decided our sample size. To underline this we have given a detailed genome data availability analysis below. We have even given the hyperlinks which can be used to check it. We have also replied to your concern about use of S. Enteritidis as reference genome. We have mentioned our decision points and reasons for this.

You have rightly mentioned about the aspect of host predilection in S. Gallinarum, which is of seminal value in bacterium’s biology. However, our work has been guided more by our vision for understanding genome of S. Gallinarum including the Indian strain from a perspective of virulence and epidemiological aspect for disease it causes, so as to pave a way for modern prevention and control measures. 

Your incisive comments have prompted us to delve deep, and we have tried to give a query by query response below. We have incorporated your suggestions into our revision manuscript. Your ideas and points were very insightful and helpful, which has not only resulted in improvement of manuscript, but also future research ideas. Thank you for your time and kind help to improve the manuscript!

regards,

1. Query

Vaid et al., conducted comparative genomics of a few (n=7) published Salmonella Gallinarum (host-restricted Salmonella serotype), S. Pullorum (another host-restricted serotype), and S. Enteritidis (a broad host range serotype) along with one strain of S. Gallinarum isolated from India and sequenced in this study. The authors then present the inventory of genetic differences and describe the results of the comparative genomic analysis of this small subset of host-adapted Salmonella serovars Gallinarum and Pullorum strains in this study. The manuscript is well written, and the analysis is appropriate. In general, it is definitely of interest to study these genomes from poultry adapted Salmonella and tease out the differences to address interesting questions related to mechanisms underlying host-adaptation, bacterial evolution, identifying markers for molecular detection, differentiation, or developing strategies for vaccine and or other control measures. However, it is unclear what was the goal of this study.

1. Response: 

The incumbent laboratory is a National microbial repository of veterinary pathogens. As Salmonella Gallinarum is an economically important pathogen of poultry in India, it was our primary interest to kickstart a comparison of first Indian S. Gallinarum genome with genomes from Asia and Latin America. Moreover, India does not import any poultry products, and it is insulated from all directions by marine and alpine physiography, so a comparison of a pristine Indian strain with other continental strain was compelling reason of this general descriptive investigation, with a focus on virulence gene subset from genome assignment. Therefore, the study aims to unravel the genomic structure and intra-serovar diversity among the eight S. enterica serovar Gallinarum strains originating from different geographic locations i.e. America, and China including Indian strain S. Gallinarum Sal40 VTCCBAA614. This has been indicated in Introduction (Line 93-107). This is vindicated by phylogenetic findings (Fig 4), in which Indian strain is distant within S. Gallinarum biovars (Line 304-307). 

Progress can be made in the direction of development of a live attenuated vaccine by a thorough understanding of the genetic virulence makeup of Indian strain. We have discussed this aspect in Discussion (Line 444-451), with obvious gains towards vaccine development. The investigation elucidated genetic structure as well as diversity amongst S. enterica serovar Gallinarum strains which can be utilized for typing, drug and vaccine targeting and surveillance. Notably, earlier studies have not included all of the currently analyzed S. Gallinarum strains (discussed in depth in next query). 

The style of our Introduction may be different although we have mentioned our goal, which is Comparative genomic comparison of Salmonella Gallinarum serovars including first Indian Salmonella Gallinarum strain genome with genomes of Salmonella Gallinarum from different geographical locations to elucidate the genomic differences and similarities with insights on virulence by analysis of Salmonella pathogenic islands, prophage analysis, TA systems, AMR genes, and their phylogeny and synteny.

2. Query:

Second, I also wondered what is new/novel finding in this study beyond what we already know about the genetic makeup of S. Gallinarum and Pullorum from multiple studies that have already been conducted on comparing genomes to demonstrate differential gene content (eg., Thompson et al., 2008; Fricke et al., 2011; Feng et al., 2013; Liu et al., 2002; Wu et al., 2005; Tang et al., 2013; Batista et al., 2018, etc)? 

2. Response: 

The esteemed learned reviewer may kindly appreciate that this is the first ever comparative study of a genome of Salmonella Gallinarum strains which also include a strain originating from Indian subcontinent. This intra-serovar study has clearly indicated the uniqueness of Indian S. Gallinarum strain by phylogenetic analysis, as well as by finding of the largest subset of singletons among all the analysed strains, which indicates its evolutionary significance. The study has revealed candidate virulence determinants which appears to be crucial for S. Gallinarum pathogenesis, such as Toxin-antitoxin systems, prophages, resistome and mobiliome, from this specific set of serovar Gallinarum strains, which has not been done earlier. 

Regarding the papers which esteemed reviewer has quoted, it is our humble submission that Thompson et al, 2008, has only studied genomic islands and prophages (not TA genes) but only in 1 strain (S. enterica Gallinarum 287/91). Fricke et al., 2011 although has included 28 Salmonella enterica strains, but these were different serovars of zoonotic significance; a single S. Gallinarum strain was included with major emphasis was on phylogeny and evolution. Feng et al., 2013 has included 4 S. Gallinarum strains however, their study mainly dealt with genomic features which lead to niche host-adaptation of host-specific pathogens. Liu et al., 2002 has studied 1 Salmonella bv. Pullorum genome and had done additional comparative study with S. Typhimurium to understand structure, evolution and plasticity of Salmonella bv. Pullorum genome. Wu et al, 2005 have studied 1 Salmonella Gallinarum strain however they have not performed WGS; rather they have prepared a physical map of S. Gallinarum and have compared it with S. Typhimurium. Tang et al., 2013 article is on Salmonella Typhi genetic content being discrete within itself as compared to Typhimurium supporting speciation. They propones that S. Typhi may be a distinct species.. Batista et al., 2018 is also not a genomic study but an experimental study on S. Gallinarum deletion mutant.

In addition to above studies, Laing et al., 2017 did a Pan genome analysis for 4939 strains of Salmonella enterica which included six S. enterica subspecies but had no representation from serovar S. Gallinarum strains in their investigation. The author’s emphasis was on pan-genome analysis from species S. enterica and other subspecies. 

(Details are also given in Response Table 1.) The Table shows that S. enterica serovar Gallinarum has been underrepresented in previous studies. 

To date, this is the only comparative study which has included highest numbers of both S. Gallinarum and S. Pullorum biovars genomes from across the world, studying their intra-serovar diversity. We agree with reviewer’s observation, that there are many studies with large number of Salmonella genomes analysed, however, these studies either have different serovars, and if S. Gallinarum has been the part of the study, they were only fewer genomes of S. Gallinarum (See Response Table 1). Furthermore, out of the multiple studies quoted, none of these have included all of the currently investigated genomes in their analysis as well as focussed on unravelling of intra-serovar diversity of specifically serovar Gallinarum. 

3. Query:

My major concern is that the study, as presented is descriptive and very limited in scope because it includes a minuscule sample size (n=8). In comparative genomics studies, the associations and inferences based on small sample size quickly fail (and become irrelevant) as soon as more data becomes available. A quick survey of the NCBI GenBank database shows that the data for there are 100s of Gallinarum and Pullorum sequenced strains exist (eg., https://www.ncbi.nlm.nih.gov/pathogens/isolates/#Gallinarum). Given this information, it makes more sense to include the sequences of all existing strains (from different geographic areas) in the comparative genomic analysis for reliably inferring pan-genomes and identifying serovar specific gene content, to address the objective of the study, as stated.

3. Response:

When we started the study we were already seized with the question of number of genomes to be included in the study (See Under Discussion 458-465). In this regard, we were guided by the paper of Rouli et al., 2015, who notes that for an open pangenome, it is more difficult to determine number of necessary strains (Rouli et al., 2015). Furthermore, NCBI assembly database at the maximum show only ~50 Salmonella Gallinarum genomes, most of which are at draft stage. 

We agree with esteemed Reviewer that inclusion of large number of samples leads to better inferences in pan genome analysis. In the present investigation, we have compared 9 genomes including 8 strains of S. enterica serovar Gallinarum and S. enterica Enteritidis str. P125109 representing different geographical reasons across the world. We considered only complete genomes for comparative investigation in order to maintain the stringency of the results as draft genomes adds bias to the pangenome analysis and can also lead to erroneous findings of candidate virulence factors. Indian strain S. enterica Gallinarum str. Sal40 was considered owing to the non-availability of complete genome of S. enterica Gallinarum from India and no representation of Indian isolate in any of the previous study. Additionally, the Indian draft genome was of good quality. The usage of complete genomes provides credence to the identified singletons and candidate virulence factors. Significantly, the currently investigated 7 S. enterica serovar Gallinarum genomes were the only complete genome sequences available in NCBI and PATRIC database when we started our investigation as well as till date. The searching of PATRIC https://www.patricbrc.org/view/GenomeList/?and(keyword(Salmonella),keyword(Gallinarum)), NCBI (https://www.ncbi.nlm.nih.gov/assembly/?term=Salmonella+Gallinarum) and NCBI (isolates browser: https://www.ncbi.nlm.nih.gov/pathogens/isolates/#gallinarum) shows 60, 43 and 53 entries for S. Gallinarum but after exclusion of draft genomes (contigs, scaffold, phage sequences), and other serovar genomes from these genomes, only seven currently investigated complete genomes are left. We emphasize the need of more completely sequenced genomes of S. enterica serovar Gallinarum from all over the world including Indian subcontinent for better epidemiological surveillance and typing. 

Given the above mentioned situation, in which majority of available genomes were at draft stage, the inclusion of such genomes would have added bias to the results as well as is not encouraged on many bioinformatic platforms for further downstream analysis. Moreover, the learned reviewer may also appreciate that our pangenome analysis is inclined towards the ‘serovar’ level, rather than Salmonella enterica ‘species’ (taxa), and that we were inclined towards finding of the subset of virulence genes in the available complete S. Gallinarum serovar genomes. 

We agree with reviewer’s excellent observation to include more number of strains for identifying serovar specific gene content, which we have mentioned in our discussion (Line 495-496). However, for this to occur, we have to obtain complete genomes of Salmonella enterica ssp. enterica sv Gallinarum bv. Gallinarum from isolates of Indian subcontinent.

The investigated genomes represent different geographical locations such as Brazil, China, USA and India. The finding of 43 new genes as depicted in the singleton development plot shows that the selected genomes shows sufficient genomic diversity and adequate representation of S. enterica serovar Gallinarum strains. 

Moreover, a number of studies have included genomes in the range of 5-10 for determination of pan-genome including the first pan-genome analysis which included eight strains of Streptococcus agalactiae, (Tettlin et al., 2005), five genomes of Legionella pneumophila (D’Auria et al., 2010); and seven genomes of Mycobacteroides spp. (Choo, Rishik, and Wee 2020).

Tettelin H et al (2005) Genome analysis of multiple pathogenic isolates of Streptococcus agalactiae: implications for the microbial “pan-genome”. Proc Natl Acad Sci U S A 102:13950–13955. https://doi.org/10.1073/pnas.0506758102

D'Auria G, Jiménez-Hernández N, Peris-Bondia F, Moya A, Latorre A. Legionella pneumophila pangenome reveals strain-specific virulence factors. BMC Genomics. 2010 Mar 17;11:181. doi: 10.1186/1471-2164-11-181. PMID: 20236513; PMCID: PMC2859405.

Rouli L, Merhej V, Fournier PE, Raoult D. The bacterial pangenome as a new tool for analysing pathogenic bacteria. New Microbes New Infect. 2015;7:72-85. Published 2015 Jun 26. doi:10.1016/j.nmni.2015.06.005

Choo SW, Rishik S, Wee WY. Comparative genome analyses of Mycobacteroides immunogenum reveals two potential novel subspecies. Microb Genom. 2020 Dec;6(12). doi: 10.1099/mgen.0.000495. Epub 2020 Dec 9. PMID: 33295861.

4. Query:

Another concern is the use of the S. Enteritidis genome as a reference here. If you want to comparative genomics of S. Gallinarum, use a reference S. Gallinarum genome and include other Salmonella such as S. Enteritidis as control/outgroup. Another approach is to use non-reference-based comparative genomics to tease out the differences using large sample sizes of Gallinarum/Pullorum strains and then scanning for serovar specificity via broader comparative genomics analysis with Salmonella enterica genomes. Unless the sample size is increased to include all available Gallinarum and Pullorum sequences in the comparative genomics analysis, the current analysis with a few strains is extremely limited and raises questions about the validity of the inferences made here, which dampens my enthusiasm for this work. I suggest that authors provide a clear hypothesis and goal of their study and focus the analysis and discussion around those goals rather than describing every difference that is detected.

4. Response: 

We used S. Enteritidis as reference genome as we were guided by previous studies. Moreover, S. Enteritidis is shown to be ancestor of Gallinarum and Pullorum (Feng et al., 2013; Hu et al., 2019; Stanley and Baquar 1994; Thomson et al., 2008). Moreover, Salmonella Enteritidis is also an important pathogen of poultry in India, and is also of zoonotic importance. In a study S. Enteritidis (30.6%) was second most frequent serotype detected in poultry after S. Gallinarum (43.7%) (Kumar et al., 2019).

Regarding the esteemed reviewer’s query on goal of the study, a response has been provided in first query above, pl.

Kumar Y, Singh V, Kumar G, Gupta NK, Tahlan AK. Serovar diversity of Salmonella among poultry. Indian J Med Res [serial online] 2019 [cited 2021 Apr 9];150:92-5. Available from: https://www.ijmr.org.in/text.asp?2019/150/1/92/268215

5. Query: 

In other words, differences exists, however, what’s important is what is the biological significance of these differences? For instance, S. Gallinarum and S. Pullorum have evolved over the years with extensive genomic degradation leading to the host adaptation [Fricke et al., (2011), Seif et al., (2018), Batista et al., (2018), etc], however, this has not been discussed at all in this paper other than briefly mentioning it in the discussion (line 440). It is important to recognize that the host adaptation is what makes S. Gallinarum so different from the rest of the Salmonella serovars. The host adaptation is complex and involves multiple factors with a significant implication on metabolic adaptation in a particular host. So simple inventory of genetic differences doesn’t necessarily mean much unless the significance of such changes is discussed with the relevant biological phenomenon. 

5. Response:

We agree with the statement that “differences exists however, what’s important is what is the biological significance of these difference”. So we have discussed the presence of differential distribution of various virulence genes. The biological significance and/or application of identified differences have been discussed for prophages (621-623) and (638-651); Toxin-Antitoxin (see Line 669-685) and AMR (725-733) More discussion on AMR genes detected has been added on Line 704-710 and 719-724.

Additionally, we have discussed the techniques that can be used in future studies to unravel the functions of identified virulence factors for their role in S. Gallinarum pathogenesis. In addition, we have thoroughly discussed virulence factors elucidated by VFDB database. We have added to the work by additionally performing BLASTN search, which has resulted in preparation of additional Table viz., Suppl.Table 7b and c. This differential distribution of Virulence factors has been added into discussion (See Line 521-572).

The evolutionary evidence towards host-adaptability of S. Gallinarum, S. Dublin, S. Cholerasuis etc host-specific serovars is overwhelmingly present in literature. Our guiding lodestar in the formation of this paper has been comparison of Indian strain of S. Gallinarum, to be compared with extant strains of different continents from the point of view of virulence factors, which has important bearing towards the development of a prevention and control strategy against the economically important disease Fowl Typhoid in India. 

Discussion on Fricke et al. (2011), Seif et al., (2018), Batista et al., (2018)

It is our humble submission to the learned reviewer that after having gone through the Fricke et al. (2011), Seif et al., (2018), Batista et al., (2018) papers, the following salient points are discerned. 

Fricke et al., (2011) compared 28 complete sequences of S. enterica strains spanning 12 serovars including S. Gallinarum 287/91 strain. The investigators compared phylogenetic, genotypic, and phenotypic data that suggested frequent loss and/or acquisition in various S. enterica sublineages and their poor correlation with the evolutionary relationships within the species. However, CRISPR locus compositions showed correlation with phylogenetic relationship to some extent in their study. There investigation indicated the potential involvement of CRISPR system in controlling horizontal gene transfer and emphasized the usage of CRISPR locus in assessment of evolutionary pattern of S. enterica sublineages for prediction and avoidance of S. enterica outbreaks. We have however not touched the CRISPR system in our analysis, hence we are hesitant to include it in our discussion. 

For Seif et al., 2018, the dataset although humungous, but is across the serovars. B Batista et al., (2018) is an experimental study on metabolic mutants of 3 genes. 

In order to elucidate the role of idnT, idnO and ccmH in S. Gallinarum str. 287/91 which are identified as conserved pseudogenes on the S. Pullorum chromosomes, Batista et al., (2018) inactivated these three genes in S. Gallinarum str. 287/91 by mutations. 

Fraikin N, Goormaghtigh F, Van Melderen L. Type II Toxin-Antitoxin Systems: Evolution and Revolutions. J Bacteriol. 2020 Mar 11;202(7):e00763-19. doi: 10.1128/JB.00763-19. PMID: 31932311; PMCID: PMC7167474.

---

## [Editor Report · Decision Letter 1]

21 Jul 2021

Comparative genome analysis of Salmonella enterica serovar Gallinarum biovars Pullorum and Gallinarum decodes strain specific genes

PONE-D-21-04830R1

Dear Dr. Vaid,

We’re pleased to inform you that your manuscript has been judged scientifically suitable for publication and will be formally accepted for publication once it meets all outstanding technical requirements.

Kind regards,

Michael H. Kogut, Ph.D.

Academic Editor

PLOS ONE
---

## [Editor Report · Acceptance letter]

2 Aug 2021

PONE-D-21-04830R1 

Comparative genome analysis of *Salmonella enterica* serovar Gallinarum biovars Pullorum and Gallinarum decodes strain specific genes 

Dear Dr. Vaid:

I'm pleased to inform you that your manuscript has been deemed suitable for publication in PLOS ONE. Congratulations! Your manuscript is now with our production department. 

Kind regards, 

on behalf of

Dr. Michael H. Kogut 

Academic Editor

PLOS ONE